# Silicification of Wood: An Overview

**George E. Mustoe** 

Geology Department, Western Washington University, Bellingham, WA 98225, USA; mustoeg@wwu.edu

**Abstract:** For many decades, wood silicification has been viewed as a relatively simple process of permineralization that occurs when silica dissolved in groundwater precipitates to fill vacant spaces within the porous tissue. The presence of specific silica minerals is commonly ascribed to diagenetic changes. The possibility of rapid silicification is inferred from evidence from modern hot springs. Extensive examination of silicified wood from worldwide localities spanning long geologic time suggests that these generalizations are not dependable. Instead, wood silicification may occur via multiple pathways, permineralization being relatively rare. Mineralization commonly involves silica precipitation in successive episodes, where changes in the geochemical environment cause various polymorphs to coexist in a single specimen. Diagenetic changes may later change the mineral composition, but for many specimens diagenesis is not the dominant process that controls mineral distribution. Rates of silicification are primarily related to dissolved silica levels and permeability of sediment that encloses buried wood. Rapid silica deposition takes place on wood in modern hot springs, but these occurrences have dissimilar physical and chemical conditions compared to those that exist in most geologic environments. The times required for silicification are variable, and cannot be described by any generalization.

**Keywords:** silicification; fossil wood; opal-A; opal-CT; chalcedony; quartz



## 1. Introduction

The abundance of petrified wood in the fossil record is not surprising, given a long evolutionary history of woody plants. The earliest known land plants appeared in the Ordovician at ~460 Ma [1]. By late Devonian, ~370 Ma, land plants had acquired most of the features of their modern descendants: roots, leaves, and woody trunks [2]. Although minerals are known to mineralize buried wood, the most important agent of wood petrifaction is silica. The reasons for this phenomenon are two-fold. Silicate minerals make up 12.6 wt.% of the Earth's crust [3], and weathering of silicate minerals provides a source of dissolved silica in natural waters. Second, the chemical properties of organic molecules in wood cause the tissue to have an affinity for precipitation of dissolved silica delivered by groundwater. This paper provides a detailed overview of the processes involved in the formation of silicified wood.

Petrifaction of wood was long been interpreted based on several generalizations. Silicified wood is commonly described as being permineralized, presuming that the original tissue is imbedded in silica minerals. Rapid rates of mineralization are inferred from observations of wood in hot spring environments. Finally, the presence of amorphous opal (opal-A), silica with incipient crystallization (opal-C/opal-CT); microcrystalline quartz (chalcedony) is considered to be the result of diagenetic transformation. Careful examination of silicified wood specimens from worldwide locations and spanning a range of geologic ages shows that, with regard to mineralization, these generalizations lack universal validity. Wood silicification may occur via a variety of geochemical pathways.

### 1.1. History of Fossil Wood Research

Humans have been aware of petrified wood since prehistoric times, but scientific investigations began in the last half-century, beginning with the 1972 paper by Buurman [4].

Within the next decade, other authors made important contributions [5–8]. These works provided a foundation for a multitude of subsequent papers that are devoted to various aspects of wood petrifaction. This paper focuses on processes involved in silicification of ancient wood, the most common form of mineralization.

### 1.2. Sources of Silica

For many silicate minerals, solubility of silica is low because of their tightly bonded lattice structure. At 25 °C at near-neutral pH, the equilibrium solubility of quartz is only 6 ppm (calculated as $SiO_2$). In contrast, the equilibrium solubility of pure amorphous silica is 70 ppm. Dissolved silica is 5–35 ppm for river water, decreasing to 2–4 ppm for ocean water, where silica is continually being removed by biologic activity and from precipitation of amorphous hydroxides of Al, Fe, Mn, Mg [9].

Silicified wood is found in environments where geologic conditions produce anomalously high levels of dissolved silica. One of these is in volcanic settings where volcanic glass is present either as tephra (volcanic ash) or as glassy matrix in lava flows [10,11]. In both instances, the amorphous silica may yield equilibrium silica concentrations of >100 ppm. Weathering of feldspar may also produce elevated silica levels. This typically occurs when feldspars are exposed to mildly acidic pH. Feldspar may be present as clasts in feldspathic sediment or as phenocrysts in crystalline rocks.

### 1.3. Dissolution of Silica

Aqueous dissolution of silica from silicate parent materials does not produce ionic silica, e.g., $Si^{+4}$. Instead, dissolution of silica in water involves a chemical reaction that produces nonionic silicic acid:

$$SiO_2 + 2H_2O \rightarrow Si(OH)_4 \tag{1}$$

### 1.4. Chemistry of Amorphous Silica

The monomeric form of silicic acid only persists in very dilute solutions ($<2 \times 10^{-3}$ M) [9]. In most natural environments, silicic acid rapidly polymerizes:

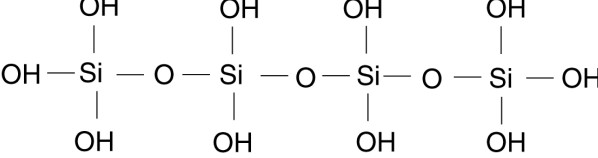

The formation of silicic acid polymers and their subsequent transformation to colloids and precipitates has received considerable attention from chemists. For a bibliography, see the 1979 monograph by Iler [9]. These reactions deserve mention because of their implications for the silicification of wood, where small differences in chemical conditions affect petrifaction.

Polysilicic acid molecules commonly have molecular weights of up to 100,000 g/mol. These polymers may aggregate to form spherical particles that have diameters less than ~50 Å. Colloidal silica develops when particle diameters exceed 50 Å. In both instances, the silica remains in suspension, so both polysilicic acid and colloidal silica are sometimes referred to as *silica sol*. The formation of silica particles involves three steps: (1) Polymerization of silicic acid monomer to form particles. (2) Growth of particles. (3) Linking of particles in a succession that begins with branched chains, proceeding to 3-dimensional networks that eventually extend throughout the liquid medium to produce a gel (Figure 1) [12,13]. These forms of silica all play important roles during silicification of wood, as described later in this report.

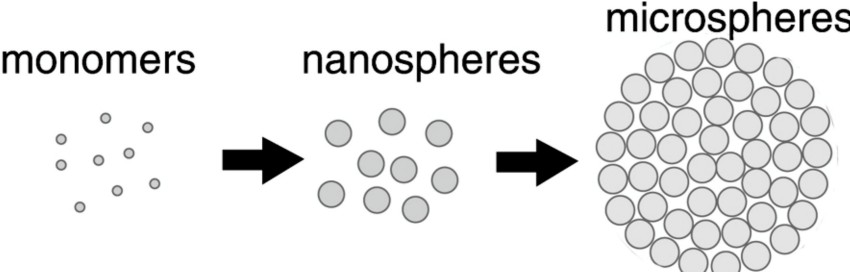

**Figure 1.** Silicic acid monomers polymerize to from microspherical aggregates. Precipitation of amorphous silica proceeds through three phases, where silicic acid polymers transition from a solution phase to a precipitated solid.

Silica may also be deposited on organic materials that have $OH^-$ functional groups, e.g., cellulosic components of cell walls. This explains onset of silicification of wood; hydrogen bonds between silica and cell materials is a preliminary step in the petrifaction process (Figure 2). Figure 3 shows an example of this process, which is known as organic templating [6].

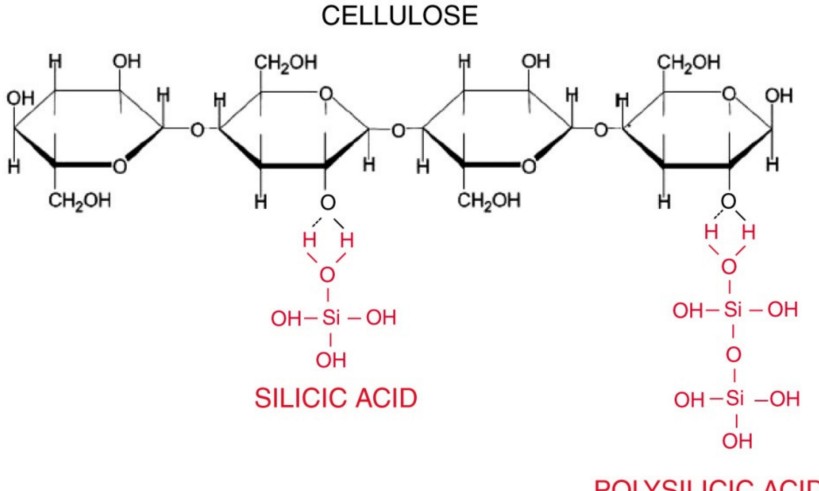

**Figure 2.** Silicic acid forms hydrogen bonds to OH groups in organic molecules like cellulose. Subsequent polymerization results from Si–O bonds. Silicic acid groups are shown in red. This silica bonding can eventually produce a silica layer on the inner surface of the cell wall.

When silica solutions are supersaturated, rapid deposition of silica may occur if a suitable substrate is present. Incipient silica deposition in cell walls via organic templating provides a substrate for further silica precipitation, resulting in development of a mineral film on cell walls. Although initial silica deposition involves hydrogen bonding, subsequent addition of silica depends on Si–O bonds.

In the absence of a nearby substrate lattice, supersaturated silica solutions produce colloidal suspensions, where polymer molecule particles provide nuclei for the condensation of additional silica. These particles these may aggregate to form silica gel. Because of their size, colloidal particles cannot pass through cell membranes, but they are an important material for the filling of open spaces, e.g., cell lumen, intercellular spaces and fractures (Figure 4).

Transitions between these silica phases involve chemical and physical complexities. When silicified wood is mineralized in multiple stages, each mineralization episode involves unique conditions, producing different forms of silica.

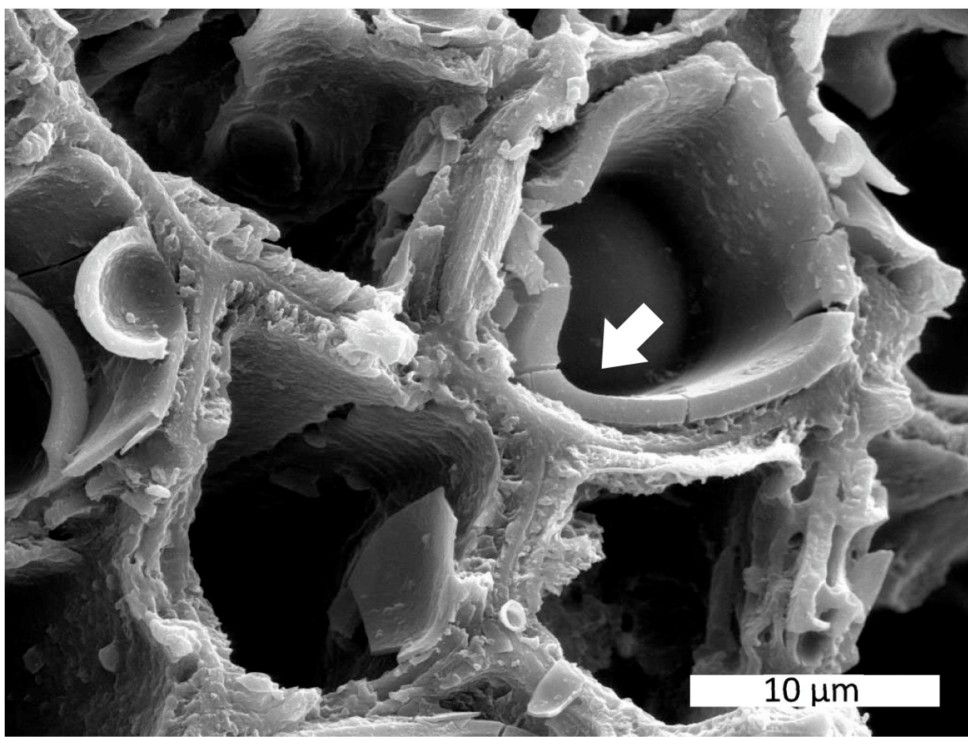

**Figure 3.** Organic templating is evidenced by silicification of multi-layered cell walls in Late Pliocene fossil wood from Red Hills Lignite Mine, Chocktaw, Missouri, USA. The multilayered architecture of the cell walls remains evident, despite the presence of precipitated silica. A continuous layer of silica on an inner cell surface shown by arrow. Specimen provided by David Lang.

Polymerization of silicic acid is affected by pH. Polymerization of monomeric silicic acid to form dimer and higher weight molecular forms is an ionic mechanism. Above pH 2, the polymerization rate is proportional to the $OH^-$ concentration; below pH 2, $H^+$ controls the rate. The growth of the initial polymer molecules is likewise affected by pH. The growing silica particles are not all the same size. Small particles are more soluble than larger ones, and the dissolution of the smaller particles provides a source of silica for the growth of larger particles. Above pH 7, the rate of dissolution and reprecipitation increases. At 25 °C, particles grow to diameters of ~5 nm, when growth rate slows. At pH below 7, terminal diameters reach 2–3 nm.

Aggregation of these small particles is also pH dependent. Above pH 6 or 7, silica particles are negatively charged, so they repel each other. The result is that particles grow in diameter without aggregation. At low pH, silica particles have very little ionic charge, so they freely aggregate to form chains, branches and ultimately 3-dimensional networks. When Si concentrations are low, at pH 2 the silicic acid monomer is largely converted to discrete particles before they begin to aggregate. At pH 5–6, monomer is rapidly converted to particles that simultaneously begin to aggregate, the two processes being impossible to separate.

In addition to pH effects, the presence of dissolved salts reduces charge repulsion, allowing aggregation to occur to produce silica gel. For fossil wood, the Si concentration of groundwater is therefore not the only important factor for wood mineralization; changes in pH and concentrations of other dissolved elements may affect petrifaction.

The conversion of sol to gel does not require an increase in silica concentration. Instead, it is involves a change in structural arrangement. Gelling occurs when about half of the silica has joined the 3-dimensional network. This gradual transformation is accompanied by an increase in viscosity, but without a large change density. As a result, wood cells may be mineralized without experiencing stress from volume changes in the silica fill.

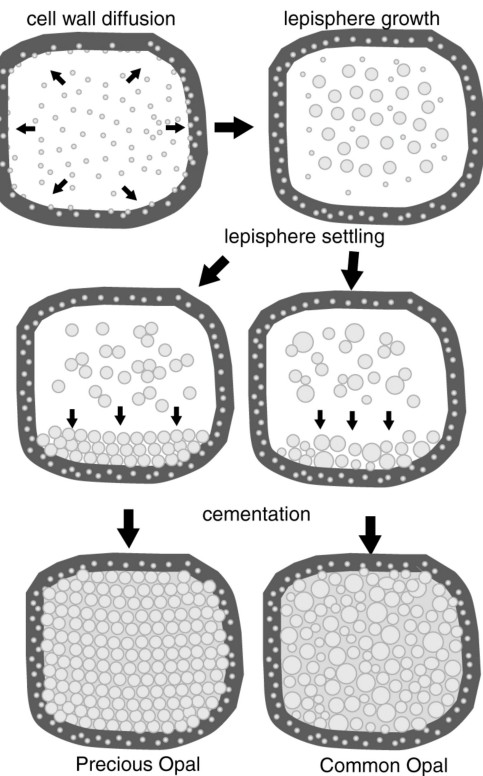

**Figure 4.** Silicification of a wood cell begins with hydrogen bonding of silicic acid monomers or small polymers to organic constituents in the cell wall. Silica in groundwater filling the cell interior (lumina) subsequently aggregates to lepispheres of varying size. These lepispheres settle by gravitation to the bottom of cell. Silica cement eventually fills interstitial spaces, producing solid opal. Depending on the degree of order in the stacked lepispheres, the final product may be either precious opal or common opal.

Amorphous opal is characterized by its relatively high water content, highly variable but commonly 4–9 wt.% [9]. The hydrous nature is the result of water in the form of silanols, where hydroxyl groups are bonded to the silica surface (Figure 5). In contrast, most of the water in crystalline opals (opal-C and opal-CT) is adsorbed $H_2O$ [14,15]. The structural nature of this water explains why many opals remain stable after exposure to atmospheric conditions. Opals of all types that fracture because of air exposure (e.g., some precious opals from Virgin Valley, Nevada, USA) owe their fragility to relatively high levels of adsorbed $H_2O$.

**Figure 5.** The water content of amorphous silica (shown in red) is predominately in the form of silanol groups shown in blue), where hydroxyl groups are structurally bonded to the silica framework.

## 2. Nomenclature of Silica Polymorphs

Among amateur fossil wood enthusiasts, silicified wood has long been divided into two categories: opalized and agatized. However, the mineral compositions of these specimens may include a variety of forms of silica. $SiO_2$ may occur in a variety of lattice arrangements. Quartz, the hexagonal version, is by far the most familiar. Three silica polymorphs are formed only under high pressure: stichovite (tetragonal), coesite (monoclinic), and keatite (tetragonal). Two forms are generally considered to have high temperature origin: tridymite (triclinic) and cristobalite. However, both minerals occur commonly as constituents of silicified wood where they formed under relatively low temperatures.

Based on X-ray diffraction patterns, Flörke [16] hypothesized that hydrous opal commonly consists of disordered intergrowths of cristobalite and tridymite, later named as opal-CT [17]. The term opal-C was added as a name for specimens that contain only cristobalite ordering [18–22]. In the absence of X-Ray diffraction data, it is usually impossible to discern the difference between opal-CT and opal-CT, and the latter term is widely used as a catch-all name for opalized woods that show incipient crystallization (e.g., in thin section optical microscopy or SEM images). Lussatite is a largely-abandoned name for a variety of opal-CT, first introduced in 1890 to describe chalcedony-like material from Lussat, France [23].

### 2.1. Amorphous Opal

The terminology of amorphous opal has evolved, with the original opal-A category divided into opal-AG and opal-AN. Detailed descriptions of silica mineral nomenclature can be found in [17–19]. Opal-An (hyalite) is typically formed in volcanic environments as a result of condensation of silica from silica-saturated vapor. This report follows the nomenclature familiar to paleobotanists: amorphous silica found in silicified wood is described as opal-A. Because of its high temperature vapor-phase origin, opal-An is not likely to occur in fossil wood.

Traces of uranium commonly cause hyalite to have bright green fluorescence (Figure 6), but ordinary opal may be fluorescent because of trace element activators.

Among gemologists, Opal-A is subdivided into precious opal ("fire opal") and potch. Precious opal is characterized by bright interference colors produced by diffraction of light from opal lepispheres that are stacked in well-ordered layers [24]. This luminescence is referred to as "play of color". Lepispheres are more randomly arranged in potch, so play of color is absent. Opal-A is usually defined as an aggregate of spherical lepispheres. However, in fossil wood, amorphous opal was also observed to occur as botryoidal forms, homogeneous layers, and aggregates (Figure 7).

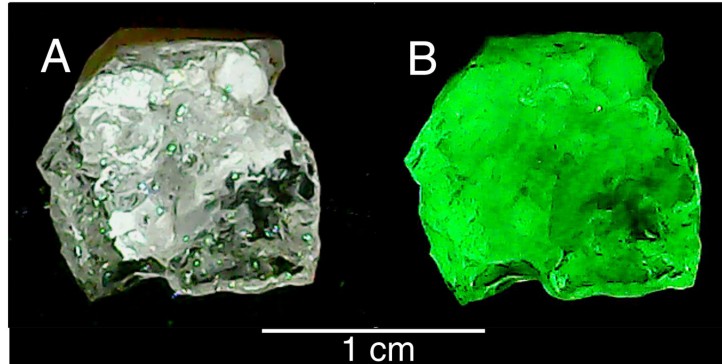

**Figure 6.** Opal-An (hyalite opal) from Zacatecas, Mexico. (**A**) Natural light. (**B**) Fluorescence at short-wave UV light (254 nm). Specimen provided by Jack Hughes.

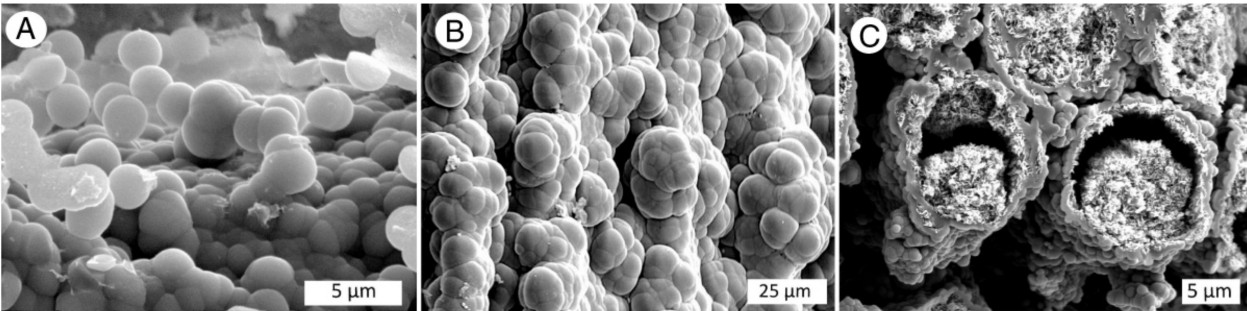

**Figure 7.** Morphologies of opal-A in Miocene wood from Nevada, USA [25]. (**A**) Lepispheres. Hazen, Churchll County. (**B**) Botryoidal texture of opal coating wood fibers. Rainbow Ridge Mine, Virgin Valley, Humboldt County (**C**). Porous aggregates filling cell lumen, with botryoidal cell walls replaced by botryoidal opal. The lumen are only partially filled, leaving crescent-shaped open spaces. Specimen from Virgin Valley, Humboldt County. Specimen provided by John Church.

### 2.2. Opal Having Incipient Crystallization

Comon opal frequently shows incipient crystallinity, where amorphous opal undergoes structural arrangement to develop lattice structures representative of cristobalite and/or tridymite. These forms, known as opal-C and opal-CT, are common in fossil wood (Figure 8).

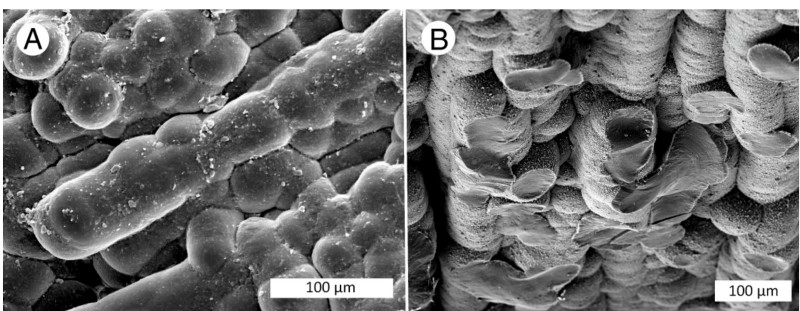

**Figure 8.** Development of incipient crystallization in Miocene fossil wood. (**A**) Wood fibers mineralized with opal-A. Gerlach, Nevada, USA. (**B**) Cells showing incipient opal-CT crystallization. The interior regions consist of vitreous opal; incipient crystallization textures are faintly visible on the outer surface of each sell, Northwest Iran, Specimen courtesy of Nasrollah Abbassi.

The paragnesis of opal-C and opal-CT is enigmatic. Tridymite and cristobalite are both commonly described as $SiO_2$ polymorphs that are found in igneous rocks. Their tendency to exist as hexagonal pseudomorphs suggests that they originate fromtransformation of quartz. The physical chemistry has been interpreted in terms of thermodynamics for high-temperature environments [26,27]. Experimental studies have been conducted at high temperatures. For example, at 1150 °C amorphous silica rapidly transforms to cristobalite [28]. However, cristobalite has long been known to occur in sedimentary rocks, where it is presumed to be a product of diagenesis [29]. Cristobalite is well-known as a diagenetic product of siliceous sinter and diatomite [30–33] and in siliceous deep-sea sediments [34]; opal-CT is the major component of common opal [21]. For silicified wood, incipiently crystallized silica has long been presumed to be the result of diagenesis of amorphous opal-A. The process is assumed to result from dissolution of opal-A, releasing silica that precipitates in situ as opal-C/CT. However, the possibility exists that opal-CT can form as a direct precipitate, based on the observation that some occurrences of opal-CT in fossil wood show no obvious morphological evidence of an amorphous precursor.

### 2.3. Microcrystalline Quartz

Among amateur collectors, silicified wood is commonly divided into two categories: opalized wood and agatized wood. As noted above, opalized wood can be mineralized with opal-A, opal-C, and opal-CT. Agatized wood is mineralized with some form of microcrystalline silica, commonly described fossil wood researchers simply as chalcedony. Mineralogists make further distinctions, using classifications that are largely based on optical mineralogy, Raman spectroscopy, electron microscopy, and X-ray diffraction. These data primarily come from forms of silica that do not include silicified wood. It is a situation where more cooperationis needed between paleobotanists and mineralogists.

The fibrous character of chalcedony may come from a variety of silica forms. Length-slow chalcedony (quartzine) is commonly a major component, an aggregate of elongate crystallites whose axis of elongation is parallel to the $\alpha$-crystallographic axis [35,36]. Length-fast chalcedony (once known as calcédonite) is a particularly common component of chalcedony, consisting of narrow twinning of right- and left-handed quartz crystals [37]. The third common component of chalcedony is moganite (also known as lutecite), a monoclinic form of silica composed of alternating layers of right- and left-handed silica lattices that lie parallel to the rhombohedral quartz surface. Moganite was originally described as a unique mineral [38,39]; it was later recognized to be associated with chalcedony [40–43].

As with opal nomenclature, for silicified wood descriptive terminology for chalcedony is inconsistent in studies of silicified wood. The issue lies in the difficulty of recognizing the presence of constituent phases. X-ray diffraction patterns show small differences when moganite is present, but Raman and FTIR spectroscopy provide clearer evidence [44]. These methods are seldom available to paleontologists, and practical recognition of chalcedony is usually based on the fibrous fan-shaped or spheroidal structures that are visible under polarized light (Figure 9).

The paragenesis of chalcedony has uncertainties. Experiments show that under heat and pressure amorphous silica may transform to chalcedony [45,46]. The limitation of laboratory experiments is that they cannot replicate the slow passage of time, and reactions are accelerated by the use of high temperatures. The results may not model conditions that occur in sedimentary environments. Fossil wood mineralized by chalcedony is exceedingly common. Do all of these specimens represent alteration of an opaline precursor? An alternate possibility is that under some conditions wood is mineralized by direct precipitation of silica, as suggested by specimens that show no evidence of a precursor phase (Figure 10A). Chalcedony has been inferred to be a primary precipitate in silicified coral [47].

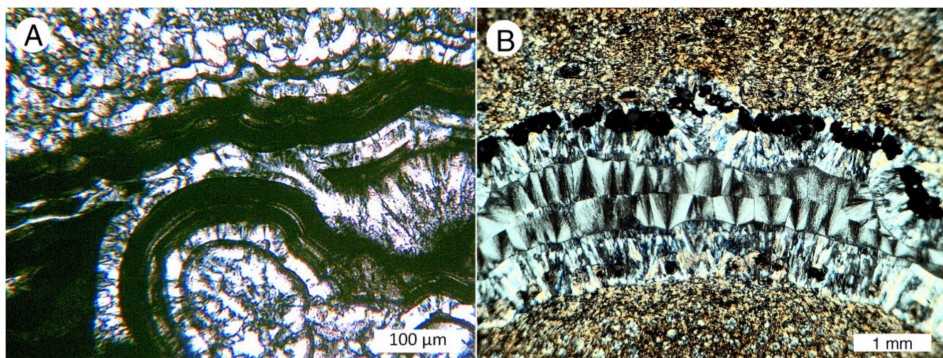

**Figure 9.** Polarized light images of chalcedony filling open spaces in fossil wood. (**A**) Miocene, Blowout Mtn., Virgin Valley area northern Nevada, USA. The dark layers are opal-A. This texture is evidence that opal-A and opal-CT were produced by cyclic precipitation, where the two polymorphs were repeatedly deposited as alternating layers. (**B**) Fracture containing chalcedony layers precipitated in two layer pairs. Both sequences show patterns of radiating fibers, with a coarse pattern present in the innermost layers. Eocene, Blue Forest locality, Eden Valley, Wyoming, USA. Specimen from Mike Viney.

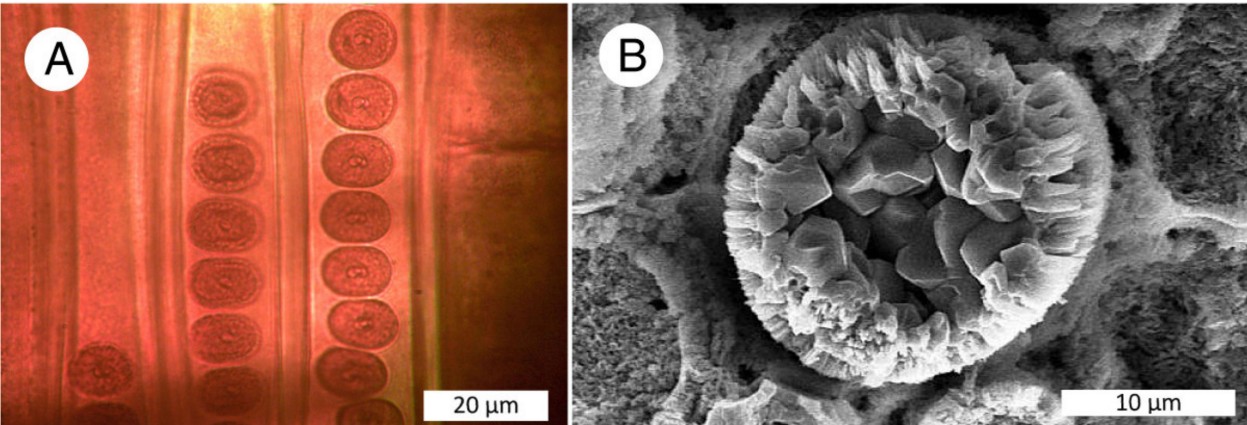

**Figure 10.** Chalcedony and quartz in fossil wood. (**A**) Chinle Formation Triassic wood, radial view showing intertracheid pits. The darker color of the silica replacing the pits is presumably due to trace element differences between the pits and adjacent cell wall. (**B**) Miocene angiosperm wood from Yakima Firing Center, central Washington, USA. A key preservation feature was that the open vessels allowed space for euhedral quartz crystals to form after the adjacent tissue had already been silicified.

Microcrystalline quartz in fossil wood can be recognized in thin sections as interlocking grains in random origin with birefringence under polarized light (Figure 11). In sedimentary environments, quartz may originate from diagenesis of an opal precursor, but quartz may precipitate directly from solution. They key requirement is the presence of a silica solution that is only weakly supersaturated, where atoms can bond at a slow rate, allowing time for the development of well-ordered lattices. Commonly, this condition occurs during late stages of mineralization, when the supply of dissolved silica has declined. This explains why quartz commonly fills vessels as a late-stage event (Figure 10B).

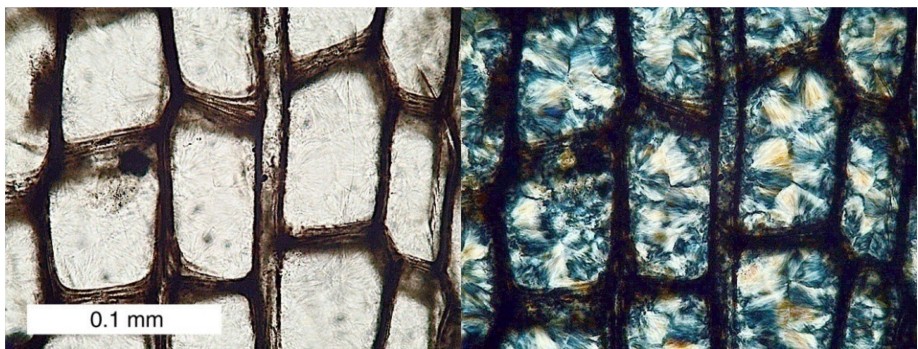

**Figure 11.** Thin section views of Carboniferous lycopod wood from the southern Rocky Mountains, Colorado, USA [48,49]. Ordinary transmitted light on right. Dark cell walls presumably owe their tint to relict organic matter. Polarized light view (**left**) shows crystalline silica filling cell lumen. Paleobotanists are seldom aware of the mineralization features of fossil wood because of their reliance on biological microscopes for observing cellular anatomy.

## 2.4. Crystalline Quartz

Megascopic quartz crystals commonly occur as a late-stage precipitate in fossil wood, where they typically occur in larger void spaces, e.g., conductive vessels in angiosperm wood and in fractures and rotted area in wood of all species. In rare instances, crystalline quartz is the primary form of mineralization for fossil wood (Figure 12).

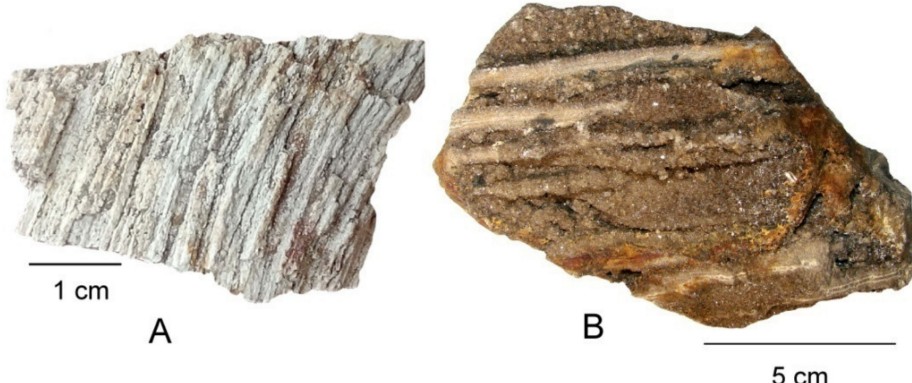

**Figure 12.** Quartz-mineralized angiosperm wood. (**A**) *Koompassioxylon*, Pleistocene, Ban Tak Fossil Forest, Thailand [50]. (**B**) *Paraphyllathoxylon*, Late Cretaceous, Brilliant, Alabama, USA. In both locations, euhedral quartz crystals have formed on the surface of silicified wood cells.

### 3. Research Methods: Field Observations

Studies of fossil wood involve acquisition of data at scales that range from geographic to microscopic. Investigations are facilitated by interdisciplinary cooperation between botanists and geologists. Taxonomic studies can be useful for estimating the geologic age of fossil wood, but for most localities age assessments are based on knowledge of regional geology and site stratigraphy. Careful mapping of the positions of individual fossil logs may provide taphonomic information. Were fossil logs preserved as upright trees? Were they transported as driftwood by a flowing river? Buried by a landslide?

Interpretations may be relatively easy when fossil logs are preserved in upright position, after burial by lahars, tephra, or lava flows. Examples include Florissant Fossil Beds National Monument, Colorado, USA, where later Eocene fossil logs represent a forest that was inundated by a single lahar flow (Figure 13A). The Eocene fossil forests at Yellowstone National Park, Wyoming, USA are more complex, because upright trunks are preserved at multiple stratigraphic levels, recording the successional replacement of forests that were episodically buried as a result of volcanic eruptions (Figure 13B). Because of its buoyancy, wood can be transported long distance by moving water, and fossil logs may ultimately be deposited at a distance from their original habitat. The largest known fossil trees were found in northwest Thailand where they are preserved horizontally in Pleistocene alluvium (Figure 13C). Careful mapping of the logs suggests that they were deposited on an alluvial fan that originated from the slopes that bordered an ancient river. The young age of the deposits allows interpretation of their origin based on modern geomorphology [50]. Paleoenvironmental reconstructions are more difficult for older deposits. Sedimentology may provide important information. For example, Thailand also contains Mesozoic fossil forests that lived long before the appearance of modern topography. In Phichit Provence, fossil logs occur in alluvium that dates from the Cretaceous, based on regional geographic mapping (Figure 14). Thin section petrography reveals that the poorly sorted sediment contains many angular clasts, characteristics that indicate high-energy deposition. Fallen trees may have been transported as part of a landslide or a flash flood. The origin remains a matter of speculation, but the sedimentology reveals that the logs were not transported by a gentle river.

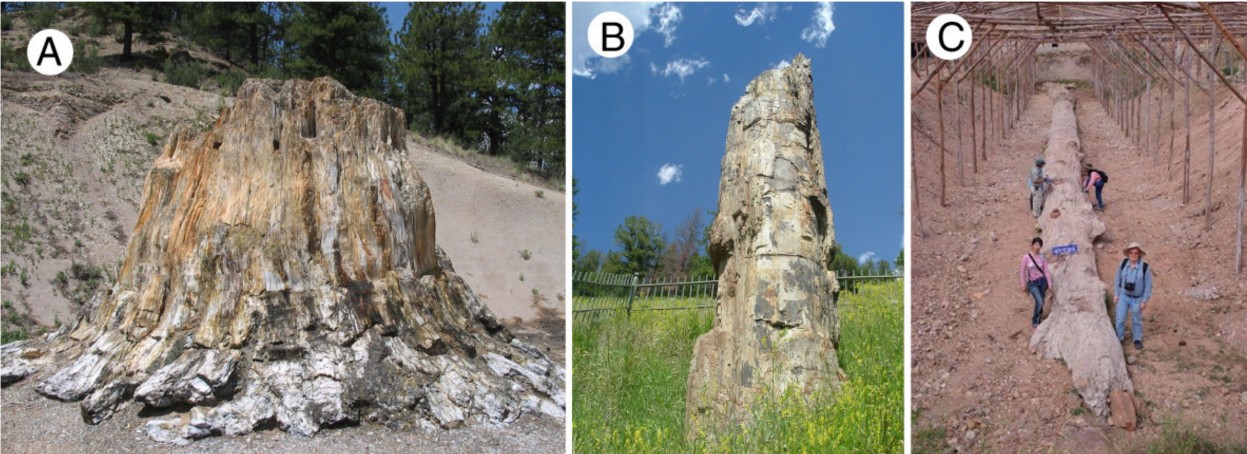

**Figure 13.** Taphonomic characteristics of fossil trunks. (**A**) In situ preservation of a *Sequioxylon* stump in a late Eocene lahar deposit, Florissant Fossil Beds National Monument, Colorado, USA [51,52]. Photo by Melissa Barton. (**B**) Upright trunk in Eocene volcaniclastic sediment at Yellowstone National Park, Wyoming, USA [53–56]. (**C**) Late Pleistocene trunk in alluvial fan deposits, Ban Tak, Thailand [50]. Photo courtesy of Nareerat Boonchai.

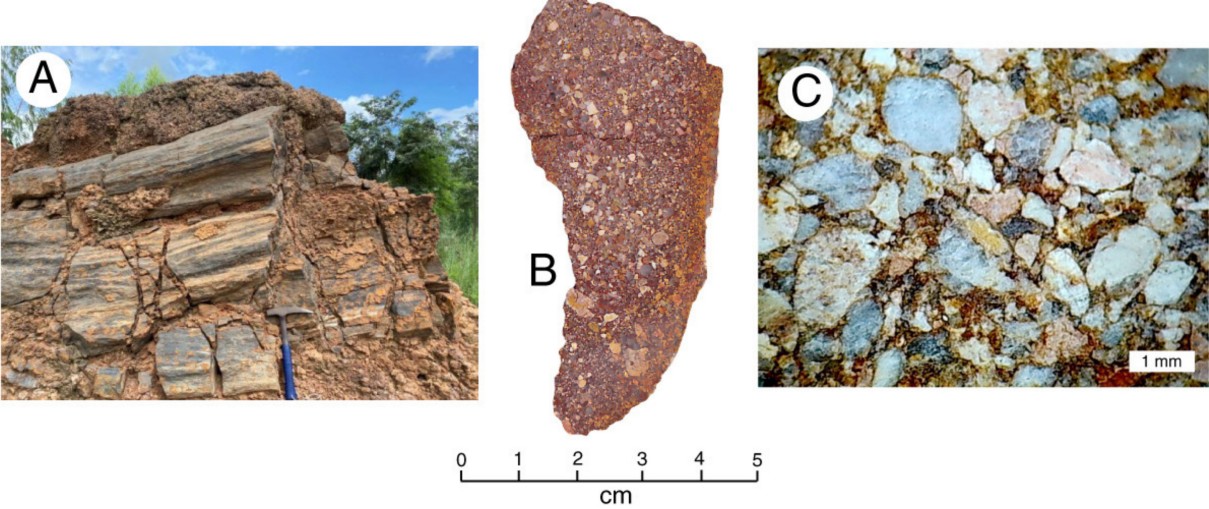

**Figure 14.** Mesozoic logs, Phichit Province, Thailand. (**A**) Logs are horizontal, enclosed by reddish alluvial sediment (**B**). Reddish color is caused by the lateritic matrix sediment. (**C**) Thin sections show angular, poorly sorted clasts set in an iron-stained matrix. Specimens provided by Yupa Thasod.

## 4. Research Methods: Observation-Based Techniques

Studies of fossil wood have traditionally relied on visual examination of specimens, optical microscopy, scanning electron microscopy, and X-ray diffraction. A variety of other analytical methods have been used by mineralogists to study silica can potentially be applied to silicified wood.

### 4.1. Megascopic Examination

Characteristics of fossil wood that can be determined by simple visual examination include tree rings, decay features, and variations in color and texture related to mineral composition (Figure 15).

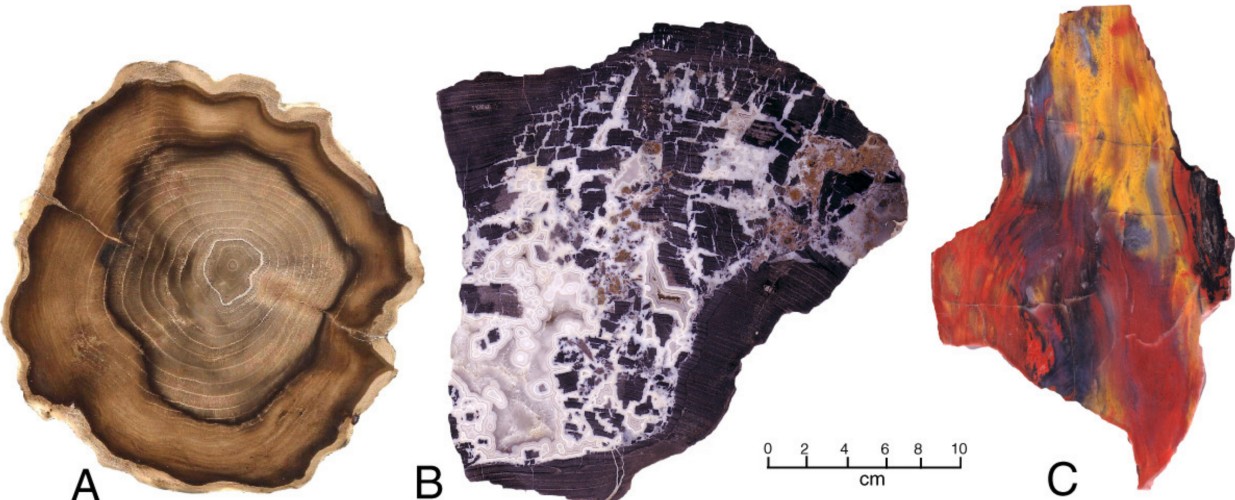

**Figure 15.** Polished slabs provide important clues for understanding the mineralization of fossil wood. (**A**) Miocene cherry wood from Hubbard Basin, northern Nevada/southern Oregon, USA. Color patterns come from variations in trace element concentrations. (**B**) Eocene wood from Tom Miner Basin, Gallatin National Forest, Montana, USA [57]. This wood had been fragmented by fungal rot prior to fossilization. Voids became filled with white chalcedony. (**C**) Triassic wood from Chinle Formation, northern Arizona, USA. The variegated colors are caused by trace amounts of iron in varying concentrations and oxidation states [58].

### 4.2. Optical Microscopy: Ordinary Light and Polarized Light

Botanical investigations of fossil wood have commonly used biological microscopes to examine cellular architecture. These images provide relatively little information about the mineral composition. Petrographic microscopes allow viewing of specimens under polarized light, where crystalline materials can be recognized based on optical interference characteristics. Polarized light microscopy can be used to identify mineral phases, and to observe the sequential order of their deposition.

For taxonomic identification, paleobotanists use slides that show three wood grain orientations: transverse, tangential, and radial. Most optical photomicrographs used in this report show transverse orientation because these views commonly provide the best look at minerals that occupy cell lumen.

### 4.3. Cathodoluminescence (CL) Microscopy

Cathodoluminescence (CL) microscopy uses visible light that is emitted by materials that have been energized by an electron beam. Samples typically consist of polished thin sections; a high vacuum environment to allow passage of electrons. CL instruments come in two varieties. The hot cathode system uses a conventional microscope that has been modified with a special specimen stage. A heated metal cathode provides a source of electrons, and the tightly sealed specimen stage is connected to a vacuum pump. Another CL method uses a scanning electron microscope with a light detector positioned near the specimen. The electron beam passes through an aperture in a curved mirror that is positioned above the specimen surface. Light emitted by the specimen is reflected by the mirror to reach the detector. Because SEM images are grayscale, rendering CL images in color requires three separate detectors, each one rendering the specimen image in a single color. These spectral signals are combined to produce a final full color image. The technical challenges make these systems very expensive, and they are limited to a small field of view. An alternate SEM method is to use a monochromatic CL detector, where images visually appear in grayscale; luminescence colors are depicted as a spectral line graph.

CL microscopy is potentially an important tool for studying the mineralization history of fossil wood, but use of the method has been limited by a lack of access to instrumentation. Most published work has come from the laboratory of Jens Götze at Institute for Mineralogy, TU Bergakademie Freiberg, Germany. Examples of CL applications include [59–67]. Figure 16 shows fossil wood images made by Professor Götze using a custom-made hot cathode optical microscope-based CL system.

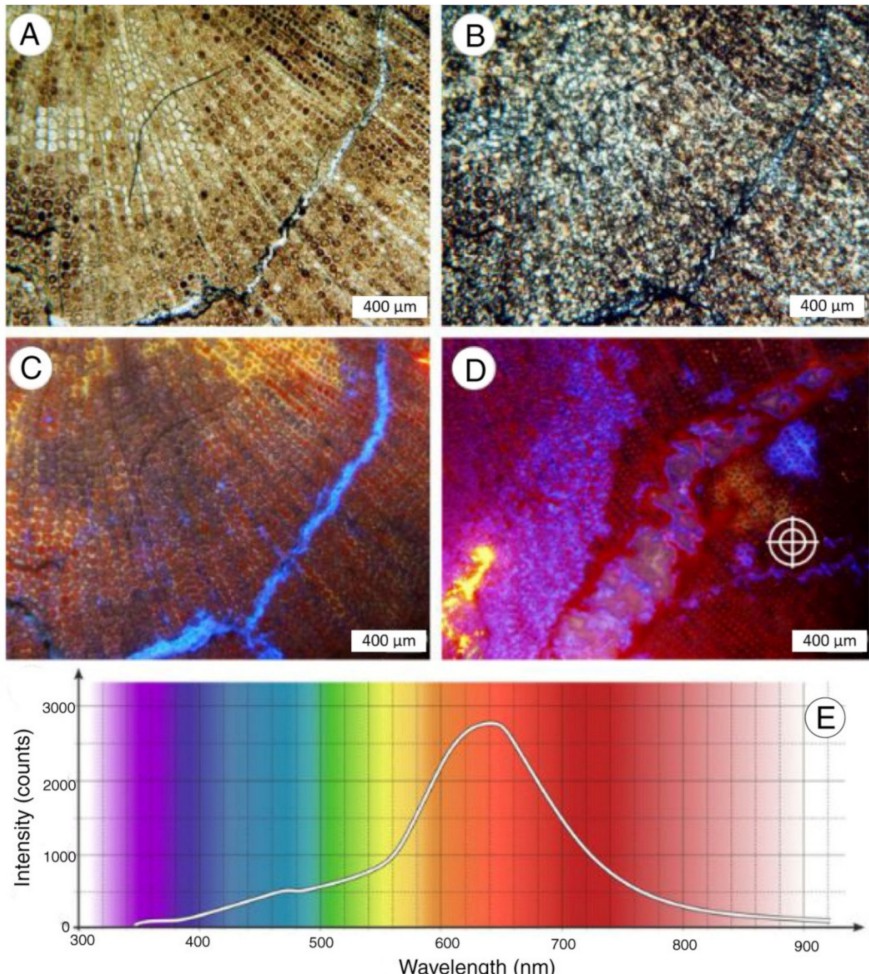

**Figure 16.** CL data from silicified wood from the Carboniferous Flöha Formation, near Falkenau, East Central Germany. This specimen is from anatomically well-preserved wood of the inner part of the root. (**A**) Transmitted light view. (**B**) Polarized light view. (**C**) CL image. Note limitation of yellow CL to the former cell walls, whereas red CL occurs within the cells. Fracture filling shows a short-lived blue CL. (**D**) All CL spectra detected from the inner root in this specimen. (**E**) Spectrum taken at marked area in (**D**) detected from the sample; image taken from the inner root. Note that anatomical preservation. Figure is modified from [67].

### 4.4. Fluorescence in UV Light

Minerals may fluoresce in response to ultraviolet light because of their inherent structure, but the most common cause of fluorescence is the presence of activator elements. For example, uranium commonly causes green fluoresce [68] (Figure 17). Reports of fluorescence in fossil wood include [69–71].

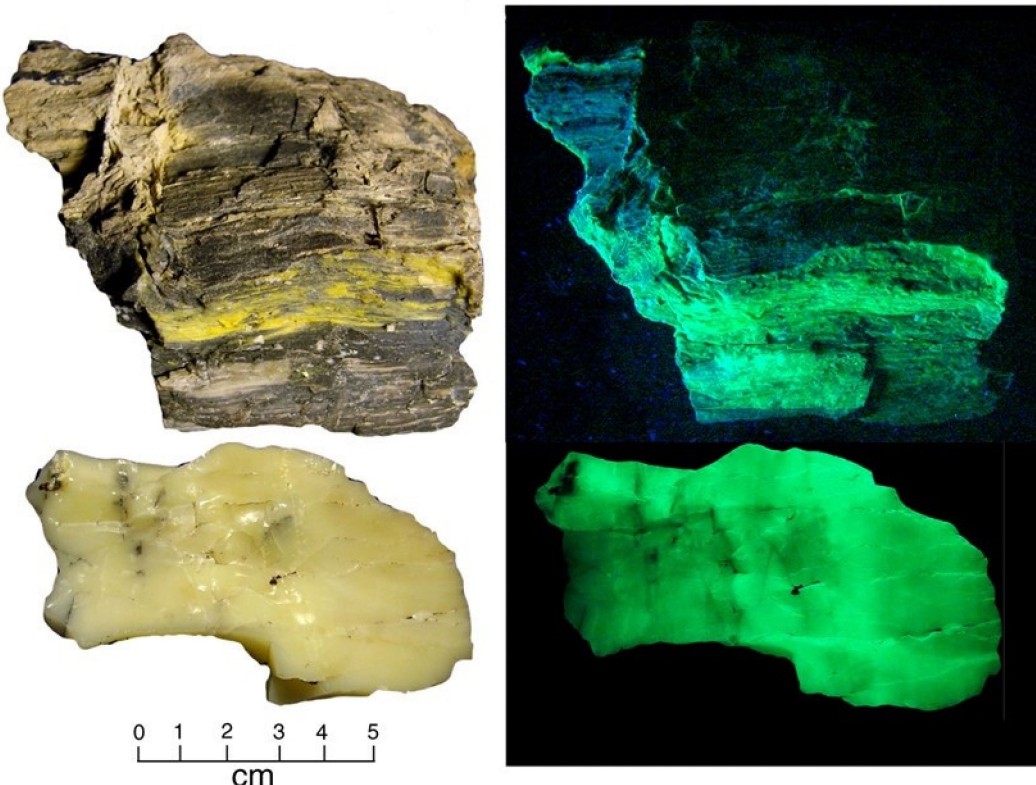

**Figure 17.** Miocene opalized wood from a uranium mining prospect in Mineral County, Nevada, USA, shows bright fluorescence under short-wave (254 nm) ultraviolet light. The upper specimen preserved wood grain texture, the lower specimen preserves no anatomical features; both were collected from the same stratum, which may represent accumulation of woody debris on the shore of an ancient lake [72].

### 4.5. Scanning Electron Microscopy (SEM)

The first uses of SEM imaging to study fossil wood came in 1972–1973 [4,73]. Electron microscopy quickly became one of the most research methods. By 1990, energy-dispersive X-ray fluorescence spectrometers were in common use, allowing semi-quantitative analysis to be done at micro scale [74]. The new generation of EDS detectors eliminated the need for liquid nitrogen chilling. One advantage of SEM/EDS is the ability to evaluate the preservation of relict organic matter based on the height of the C K$\alpha$ peak (Figure 18). Quantitative values calculated for carbon are unreliable because of the difficulty of measuring the very low energy photons emitted by this element.

Although transverse views are preferred for optical microscopy, this orientation is less commonly used for SEM imaging when specimens are prepared as fractured blocks. In this instance, many silicified woods tend to split parallel to the elongate cells, so SEM mounts are likely to show either radial or tangential orientations. SEM images primarily show topographic features, and the best fossil wood images views are obtained with specimens that have incompletely mineralized intercellular spaces and/or cell lumen.

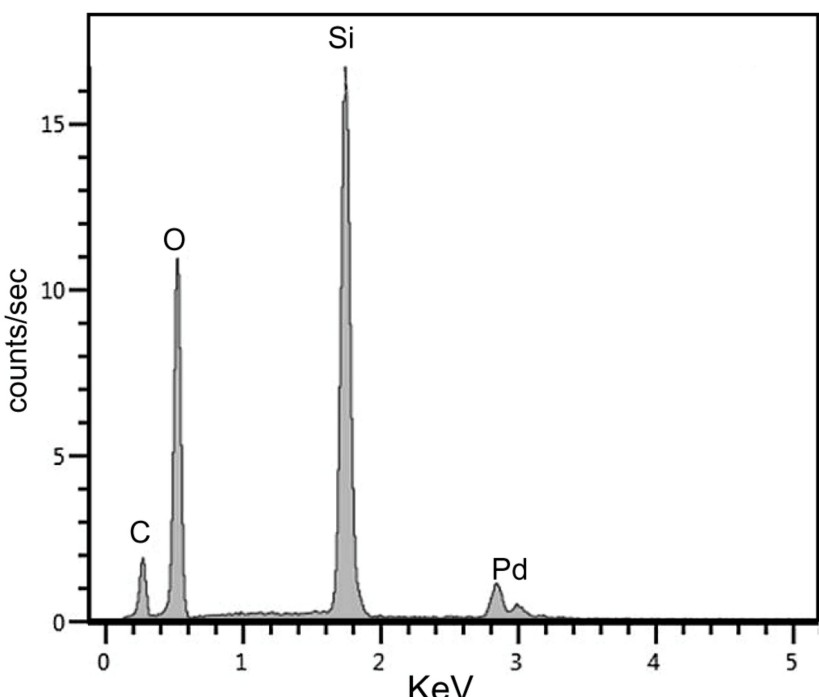

**Figure 18.** SEM/EDS spectrum of Miocene silicified wood from Meshgin Shar area, northwest Iran [75]. The small carbon peak is evidence that very little of the original organic material was preserved during fossilization. The Pd peak is from a thin palladium coating applied to the specimen surface to provide electrical conductivity.

## 5. Research Methods: Analyses to Discern Mineralogy

### 5.1. Density and Loss on Ignition

Measurement of density (specific gravity) is a simple method for determining the mineralogy of silicified wood. Characterization is based on the observation that opal wood densities are typically in the range of 1.9–2.1 g/cm$^3$, and woods mineralized with chalcedony or quartz have densities of 2.3–2.6 g/cm$^3$ [76]. Exceptions occur when fossil woods contain open spaces, large amounts of organic matter, or non-silica minerals. For woods mineralized with chalcedony or quartz, loss on ignition (LOI) at 450 °C provides a method for quantifying the amount of relict organic matter. The calculation is based on the mass loss after heating, compared to the density of the wood prior to fossilization. The latter value is estimated based on data from modern trees. The LOI method is less accurate for opal wood, because dehydration of the hydrous silica contributes to the change in mass.

### 5.2. X-ray Diffraction (XRD)

The elastic scattering of X-rays by crystalline materials provides a method for determining their lattice structure. The method has long been used to study the composition of fossil wood, and XRD is the primary method for recognizing the presence of various silica polymorphs (Figure 19). XRD has several attractive characteristics. Analyses are relatively easy to perform, requiring only a small amount of powdered sample. Modern diffractometers use rapid computer methods for the identification of components. Disadvantages include the high costs of instruments, and the inability of detecting non-crystalline materials. For fossil wood, amorphous phases may include relict organic matter and amorphous or weakly crystallized phases. Leisegang et al. [77] provide a detailed overview of the use of XRD for fossil wood analysis. Examples include [4,5,7,8,25,58,65,78–81].

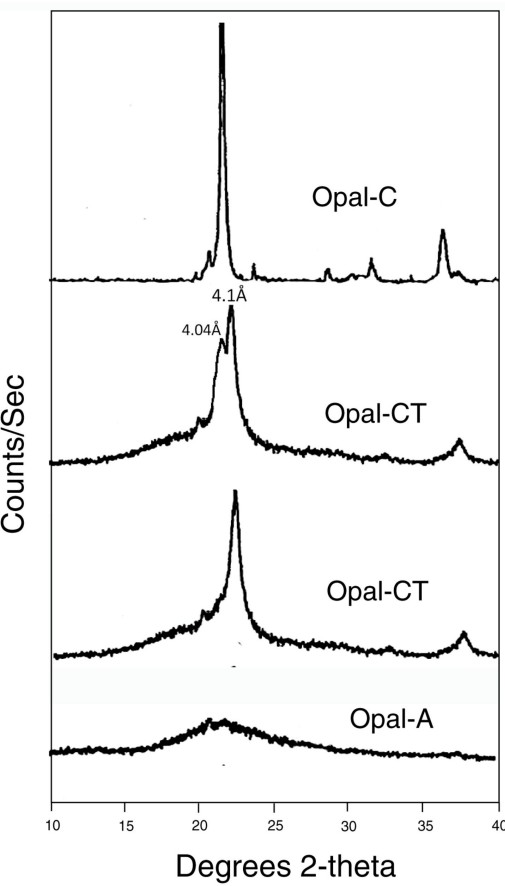

**Figure 19.** X-ray diffraction patterns for various forms of opal, using Cu-α radiation. Adapted from [77].

*5.3. FTIR/Raman Spectroscopy*

These spectroscopic methods can be used to identify silica polymorphs present in petrified wood (Figure 20).

Fourier Transform Infrared (FTIR) spectroscopy uses modulated, mid-infrared energy to investigate bond energies within a sample. The absorption of infrared light at specific frequencies is directly related to the atom-to-atom vibrational bond energies in the molecules. When the bond energy of the vibration and the energy of mid-infrared light are equivalent, the bond can absorb that energy. Various bonds in a molecule vibrate at different energies, causing them to absorb different wavelengths of the IR radiation. The frequencies and intensities of these absorption bands produce a spectrum that is a fingerprint for the molecule. Modern FTIR instruments analyze all wavelengths simultaneously. The may be free-standing instrument, or combined with the optical system of a microscope. IR Spectroscopy has been used to study opal and other forms of silica [82–86].

Raman spectroscopy uses light scattering to analyze chemical composition. As visible or near-infrared light interacts with molecular vibrations, the light becomes scattered, an energy loss that causes an increase in the light's wavelength. The changes of wavelength are very specific to a molecular vibration and plotting changes in wavelength as a spectrum produces a "molecular fingerprint" that can be used to identify and quantify chemical composition. Raman spectroscopy can also provide information about crystal lattice and molecular backbone structure. Research examples include [64,77,87–90]. Applications of Raman spectroscopy for paleontology research have recently been reviewed [90].

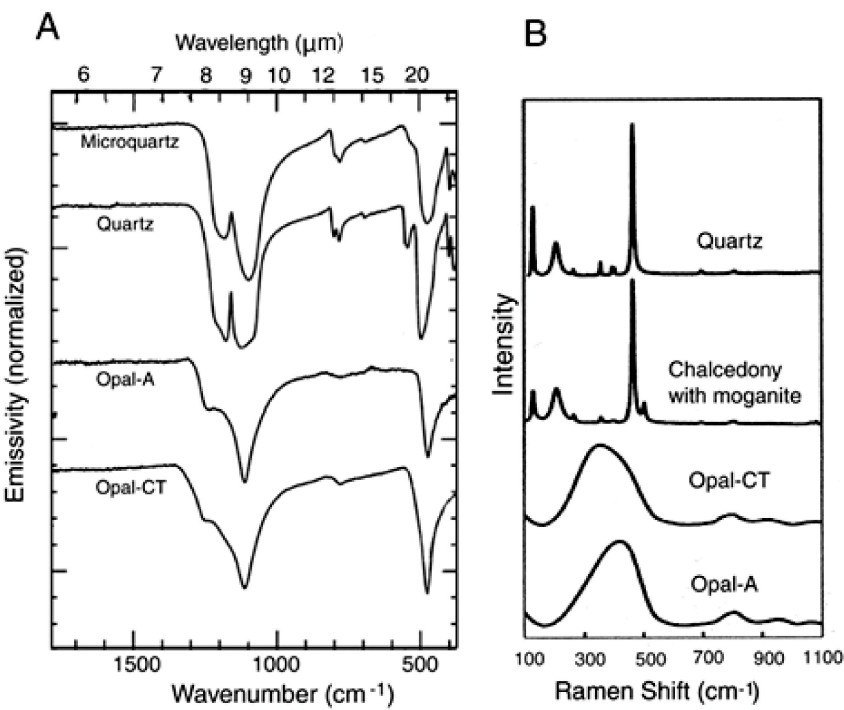

**Figure 20.** Examples of FTIR and Raman spectra for silica polymorphs. (**A**) FTIR spectra of silica minerals [83]. (**B**) Raman map of silica phases in silicified wood from the Morrison Formation, Escalante State Park, Utah, USA [86].

*5.4. Nuclear Magnetic Resonance (NMR) Spectroscopy*

During NMR spectroscopy, samples that are placed within a strong magnetic field are subjected to radio frequency waves. Under these conditions, the nuclei of atoms may show transitions in spin state that are specific to the particular isotope and its environment. These energy level fluctuations, known as "resonance", are detected as shifts in the radio frequencies. NMR spectroscopy can only be used on nuclei that have spin. The most commonly used NMR isotopes are [1]H and [13]C, but most elements have at least one isotope that is NMR active. [29]Si is a suitable NMR element, so the method can be used to study the chemical structure of silicate minerals [91–93].

## 6. Analyses to Characterize Elemental Composition

*6.1. Laser Ablation Inductively Coupled Plasma Mass Spectrometry (LA-ICP-MS)*

ICP-MS spectroscopy involves the use of a high-frequency plasma torch to ionize molecules to produce vapor that is then transported by an inert carrier gas to the strong magnetic fields of a mass spectrometer. Individual ions are dispersed in an arc based on their mass and electrical charge. By using calibration standards, the composition of unknown samples can be quantitatively determined with high precision, including trace elements. The laser ablation method is useful for geologic materials, because the laser beam can volatilize elements from solid samples, eliminating the need for chemical dissolution of samples. The sample area that is energized by the laser can range from a single tiny spot to lines with lengths of several mm (Figure 21). Geological applications of LA-ICP-MS have been presented by [58,94,95]. Because the mass spectrometer detects atoms based on their mass and electrical charge, the ICP-MS method can be used to determine U-PB radiometric ages for geologic samples, including fossil wood [96]. Like CL microscopy, the major limitation for the paleontology use of LA-ICP-MS is the lack of access to facilities.

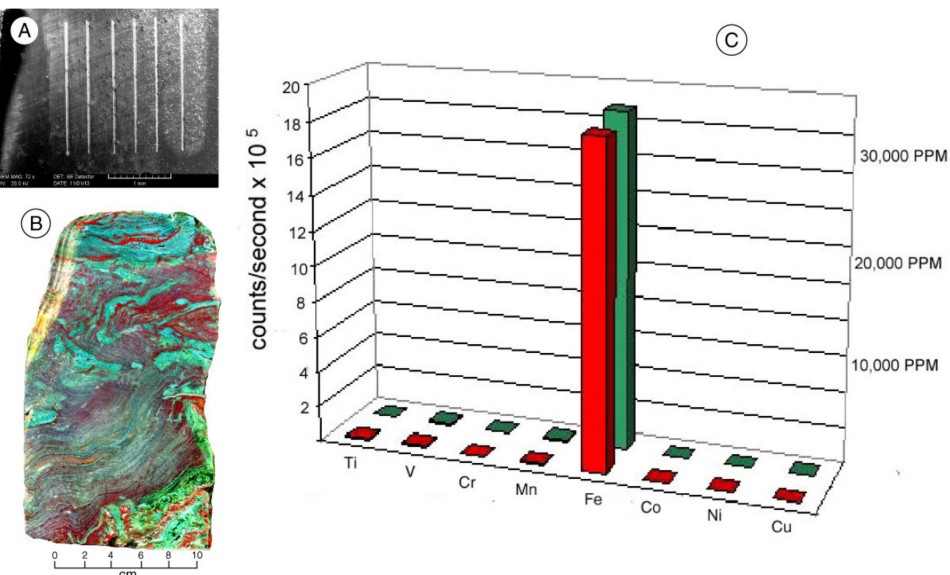

**Figure 21.** ICP-MS analysis of elements released from Eocene silicified wood from Hampton Butte, Oregon, USA, reveals trace elements abundances. (**A**) SEM image of sample after analysis. Each line is 2 mm long. For each sample, data from six lines were averaged. (**B**) Red and green colors in this Hampton Butte slab are both caused by iron in differing oxidation states [58]. The swirling patterns originated during diagenesis, and do not represent anatomical characteristics of the original wood. (**C**) Histograms showing trace element abundances. Red and green wood colors are both caused by iron, with different oxidation states producing both hues.

*6.2. X-ray Fluorescence Spectroscopy (XRF)*

X-ray fluorescence spectroscopy provides a powerful tool for analyzing the elemental composition of geologic materials, including fossil wood. The method is based on the principle that when atoms receive energy from an outside source, electrons maybe raised to higher orbital levels. When an electron returns to a lower orbital, the energy is released in the form of an X-ray photon. The energy of this photon is equal to the energy difference between the two orbitals. This quantum-controlled emission means that X-ray fluorescence can be used to recognize the presence of particular elements, and determine their abundance. XRF spectrometers can be divided into several categories. Microbeam methods use a focused beam of electrons to excite the sample, allowing analyses to be performed on very tiny regions. Electron microprobe analysis (EPMA) of specimens prepared as polished thin sections provides high-precision measurements of major and minor elements. Scanning electron microscopes (SEM) are commonly equipped with energy-dispersive X-ray detectors, allowing them to be used for microbeam analysis (Figure 18). The method, commonly referred to as SEM/EDS, can be used on virtually any specimen, but the quantitative results are less accurate than EPMA data. Another approach is to use bulk XRF analysis, where the entire surface of a sample is irradiated in order to determination of the average composition. The energy source is typically a broadly focused X-ray tube. Detection of low-energy X-rays emitted by light elements (e.g., C, O, Na, Mg, Si, Al) requires that the specimen chamber be evacuated because these photons will not penetrate air. Geologic specimens are typically prepared as pressed powders, or as borosilicate glass discs made by fusion of rock powder with a lithium borate flux at 1000 °C. Portable XRF spectrometers use a radioisotope as an energy source. Their small size and user-friendly software make them popular for applications such as mineral prospecting, ore processing and metallurgy. However, their poor sensitivities for Si and Al and their inability to recognize C and O limit their usefulness for analyzing fossil wood. Details of the XRF methods can be found in [97–99].

## 7. Laboratory Methods: Determination of Water Content

*Thermal Analysis*

Several methods can be used to study compositional changes that occur when a sample is gradually heated. For silica minerals, these changes are primarily related to loss of adsorbed or structural water. Thermal analysis has been an important source of information for understanding the physical chemistry of hydrous silicates (e.g., opal), but the method has so far had only limited use for studying fossil wood.

Thermogravimetric analysis (TGA) is performed by heating a small specimen in air or an inert gas atmosphere, precisely measuring the change in mass as the temperature rises. Absorbed water will evaporate by around 100 °C. Structural water in hydrous silicates will evaporate by around 400 °C, so the degree of hydration may be determined. When samples are heated in air, mass may decrease because of combustion of organic matter, or increase from oxidation of elements like iron. Heating in nitrogen avoids these reactions.

Differential thermal analysis (DTA) measures the heat input required to increase a specimen's temperature. Values are calculated with respect to a standard reference material, hence the "differential" terminology.

Examples of use of thermal analysis include [14,100].

## 8. Wood Silicification Processes

Fossilization is popularly conceived as being a very slow process, but time alone is not the most important factor. For wood, the botanical characteristics favor mineralization. Stem tissue provides structural support of an upright trunk, but hollow xylem cells provide fluid transport. After death, these cells allow the permeation of mineral-bearing groundwater. However, groundwater can only reach the buried trunk if the surrounding sediments are permeable. Wood in impermeable (e.g., clay-rich) sediment may remain unfossilized even after tens of millions of years, as evidenced by major sites in the Canadian Arctic and Europe [101].

### 8.1. Tissue Degradation versus Mineralization

Mineralization occurs as a dynamic balance between tissue degradation and silica deposition. Wood tissue is primarily composed of cellulose and lignin, which have differing resistances to degradation. Because cellulose decomposes by hydrolysis and oxidation at a much faster rate than lignin, it is likely to be replaced by silica before the lignin is degraded. Cellulose provides the structural framework for wood cells, and in fossil wood anatomical preservation depends on precipitation of silica contemporaneous with the loss of cellulose. The final result is an inorganic pseudomorph of the original tissue.

The persistence of lignin may explain the dark color that is commonly observed in silicified cell walls, where carbonaceous matter is a relic of the original organic components (Figure 22).

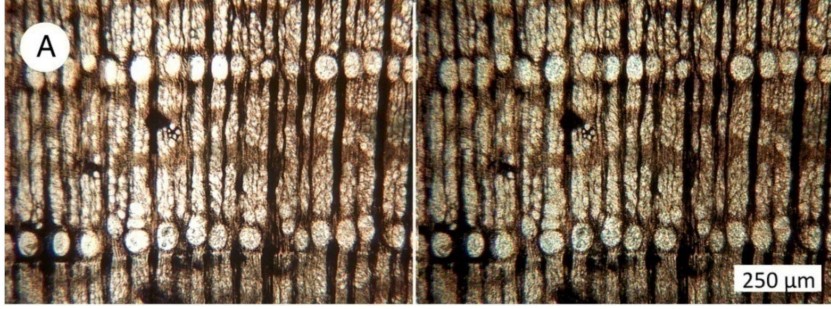

**Figure 22.** *Cont*.

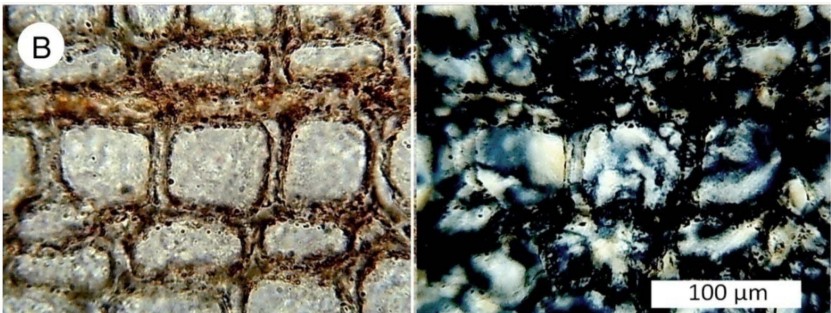

**Figure 22.** Fossil wood showing dark cell walls that are presumably caused by relict organic matter. transmitted light view on left, polarized light illumination on right. (**A**) Miocene angiosperm wood from Goose Creek, Nevada, USA. (**B**) Cretaceous gymnosperm wood, Phichit, Thailand. Polarized light views show the mineralization. Note that the angiosperm wood features ovoid vessels surrounded by small-diameter cells, in contrast to the rectangular cross-section of tracheid cells in gymnosperm wood.

### 8.2. Permineralization

Petrifaction of wood has long been described as a process of permineralization, where plant tissue becomes entombed by minerals. The assumption is that if these minerals could be dissolved, the original tissue would be revealed. An alternate interpretation is that minerals replace the original organic matter. Silica permineralization can be demonstrated under laboratory conditions [102], and several authors have interpreted features in fossil wood as evidence of void filling [103,104]. However, the popularization of the permineralization hypothesis has evolved despite a paucity of supporting evidence. Little attention has been paid to a 1927 study where silicified wood specimens were treated with HF to remove the silica [105]. The results showed that most specimens contained very little of the original organic matter. More recent testing confirms this discovery [106]; for most silicified woods, only a small percentage of relict organic matter is preserved. Permineralization is a common phenomenon in woods mineralized with calcium carbonate [107], but rare in silicified wood.

## 9. Examples of Wood Silicification
### 9.1. Opal-A

Opal-A primarily occurs in Neogene fossil wood, but most opalized woods are mineralized with opal-C or opal-CT. The latter minerals can be distinguished in bulk samples by XRD, but they cannot be distinguished by ordinary microscopy. However, opal-A can be recognized in polarized light optical images because of its isotropic nature, in contrast to the low birefringence characteristic of opal-C/CT (Figure 23). In SEM images, opal-A may be visible as smooth-surfaced lepispheres (Figure 24) and as botryoidal vitreous masses and porous aggregates (Figure 25).

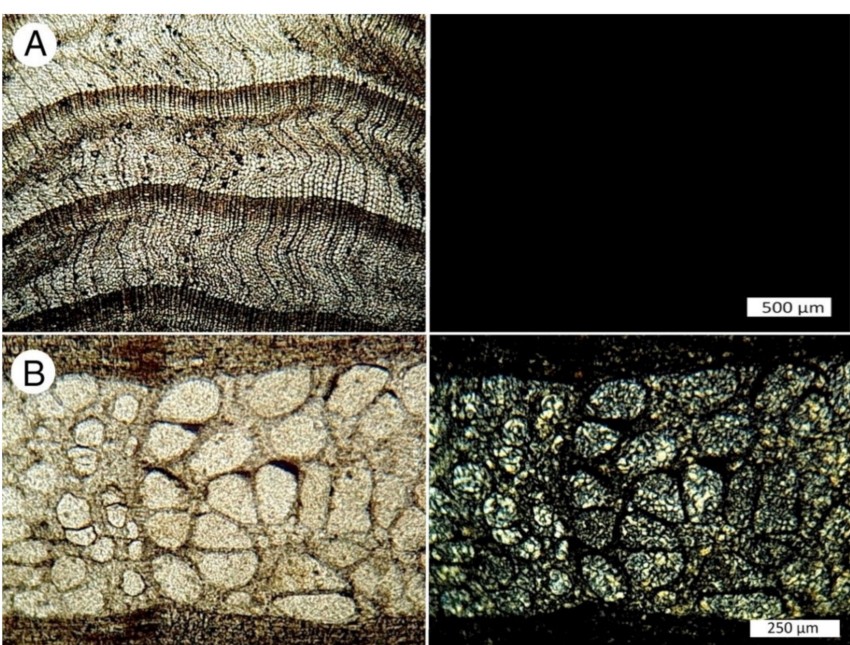

**Figure 23.** Petrographic microscope views of woods mineralized with opal-A and opal-CT. Ordinary transmitted light view on left, polarized light image on right. (**A**) Transverse view of Miocene wood from Georgia nation. The isotropic nature of the polarized view is evidence of amorphous opal-A. Note the pronounced difference of cell sizes that comprise the three pairs of annual growth rings. Specimen provided by Miriam Makazde. (**B**) Opalized Miocene wood from Badger Pocket, Yakima Firing Center, central Washington, USA. Polarized light view shows the low birefringence typical of opal-CT that fills cell lumen of cells within a single growth ring. Specimen provided by T.A. Dillhoff.

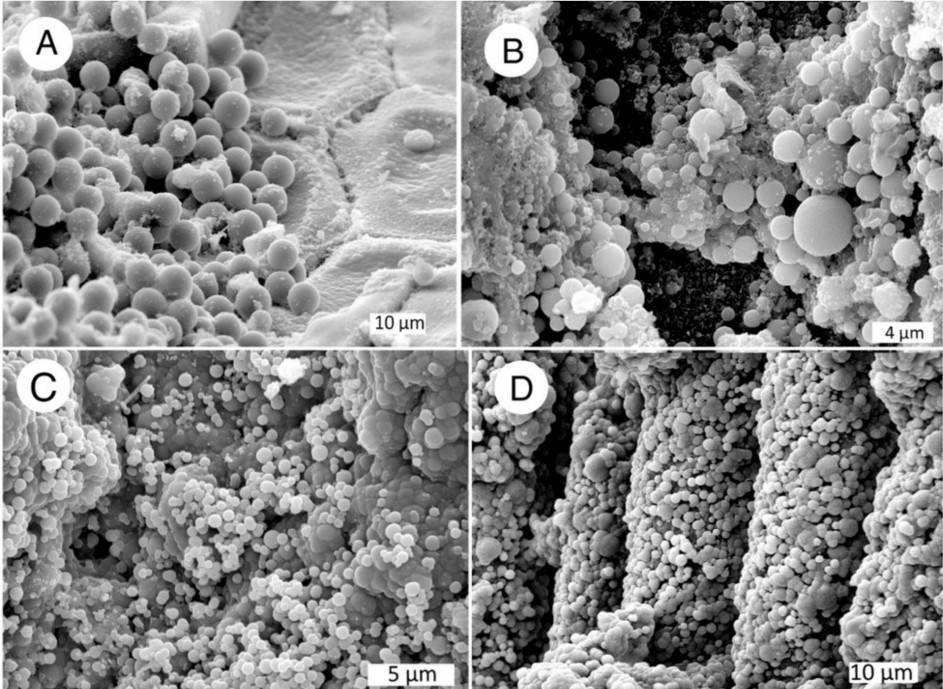

**Figure 24.** *Cont.*

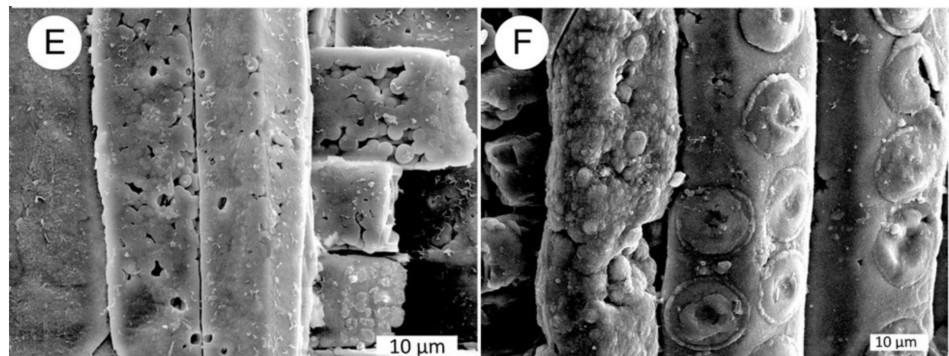

**Figure 24.** Opal-A in fossil wood from Miocene locations in Nevada, USA, showing lepisphere architecture. (**A**) Loosely-packed lepispheres lying on silicified wood cells, seen in transverse orientation. Cecilly Ann Claim, Virgin Valley, Humbldt County. (**B**) Schurz, Mineral County. (**C,D**) Tracheid cells replaced by opal-A lepispheres. Virgin Valley, Humboldt County. (**E**) Hazen, Churchill County. The perpendicular angles of the two sets of cells are typical of wood cut in a tangential orientation. The horizontal cells are "rays", which increase the mechanical strength of the wood, reducing the risk of vertical splitting caused by the parallel alignment of the vertical tracheid cells. (**F**) Middlegate, Churchill County. The prominent circular "bordered pits" are characteristic of gymnosperm wood. Specimens (**A,C,D**) provided by John Church.

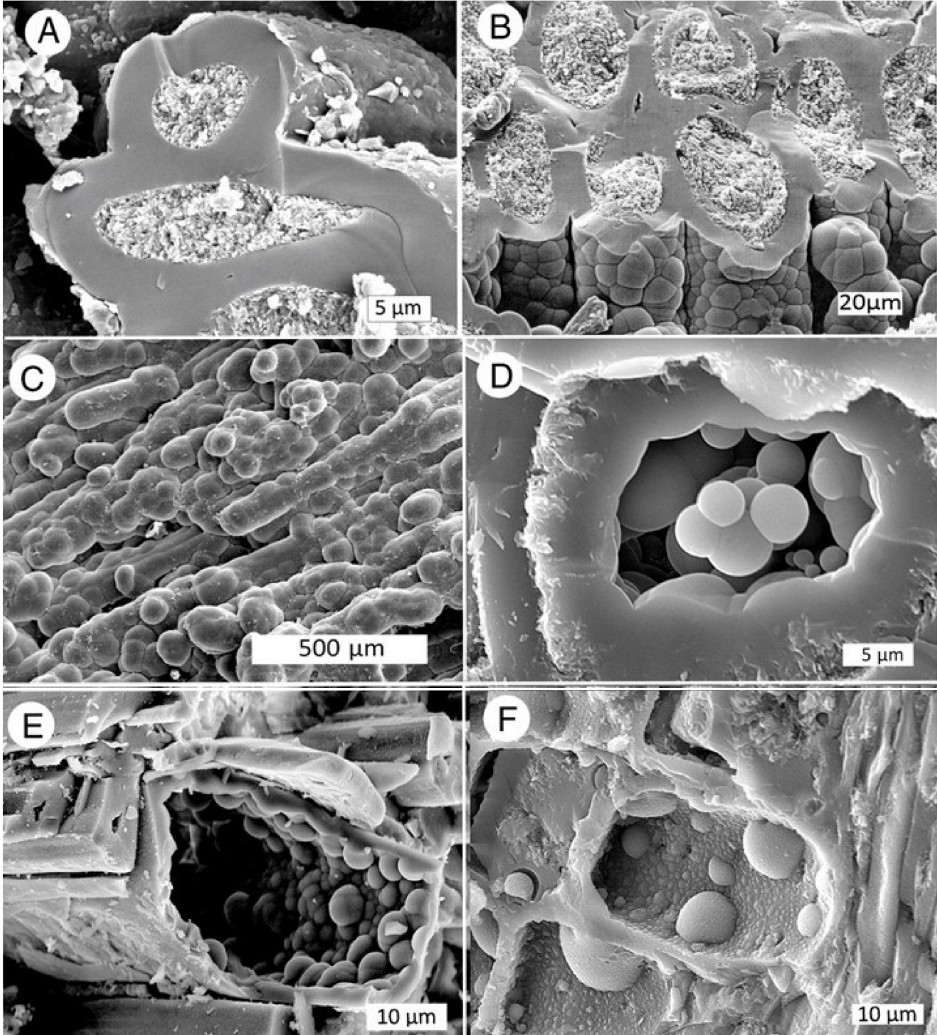

**Figure 25.** Botryoidal opal-A from Miocene locations.(**A–E**) are from Nevada, USA. (**A,B**) Oblique

transverse views of wood from Northern Washoe County. Cell walls are mineralized with botryoidal opal-A, lumen contain porous fillings (**C**) Longitudinal view shows cells coated with botryoidal opal, with intercellular spaces mostly unmineralized. Gerlach, Washoe County. (**D**) Transverse view of a single cell showing botryoidal cell walls and lepispheres partially filling lumina, northern Washoe County. (**E**) Conifer wood tracheid with relatively thin opalized cell walls with hempispherical coating interior surfaces. Lyon County. (**F**) Cells containing sparsely distributed hemispherical masses on opalized cell walls. Georgia nation specimen provided by Miran Makazde. These photomicrographs show that the first stage of silicification is typically the mineralization of the cell wall, leaving interior regions (lumen) that may or may not become mineralized.

*9.2. Opal-C/CT*

Most opalized wood consists of opal having incipient crystallinity. SEM images show cell wall surfaces continuously covered by bladed crystals (Figure 26). Bladed lepispheres are typically found in open spaces where the aggregates had room to develop (Figure 27). In confined spaces, crystallinity may not be evident (e.g., Figure 26C).

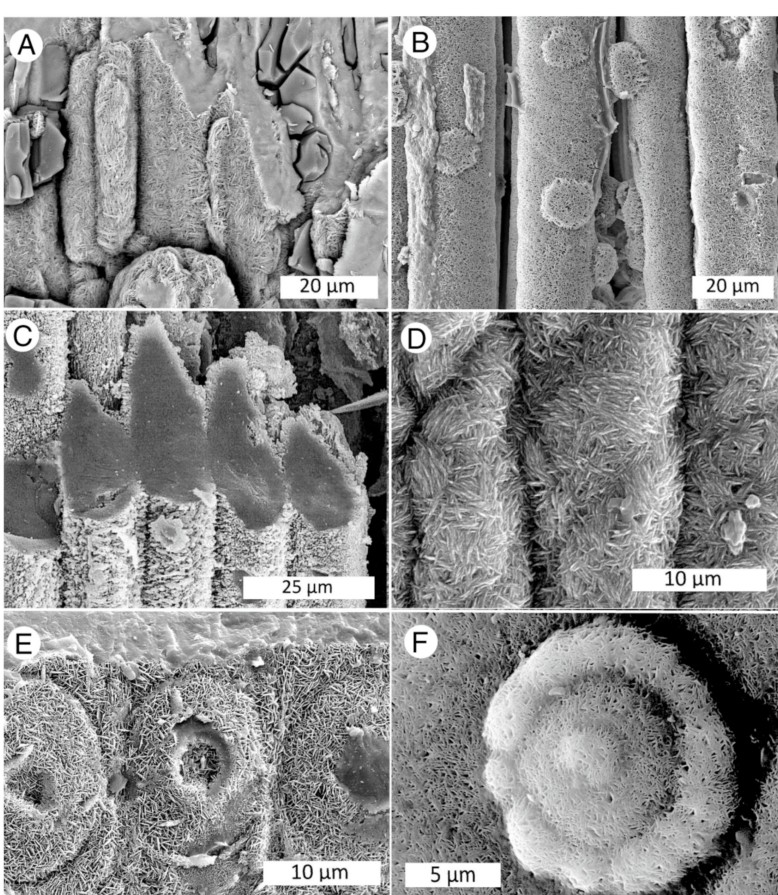

**Figure 26.** Longitudinal views of Miocene woods mineralized with opal-CT as continuous surfaces. (**A**) Northwest Iran. (**B**) Vantage, Washington, USA. (**C**) Mehrten Formation Sierra Nevada Mountains, California, USA. Note that in this wood the crystal texture is only evident on the exterior surfaces of cell walls. (**D**) Clover Creek, Idaho, USA. (**E**). Conifer wood from Owyee Desert, Oregon, USA showing preservation of bordered pits. (**F**) Single bordered pit, Middlegate, Nevada, USA.

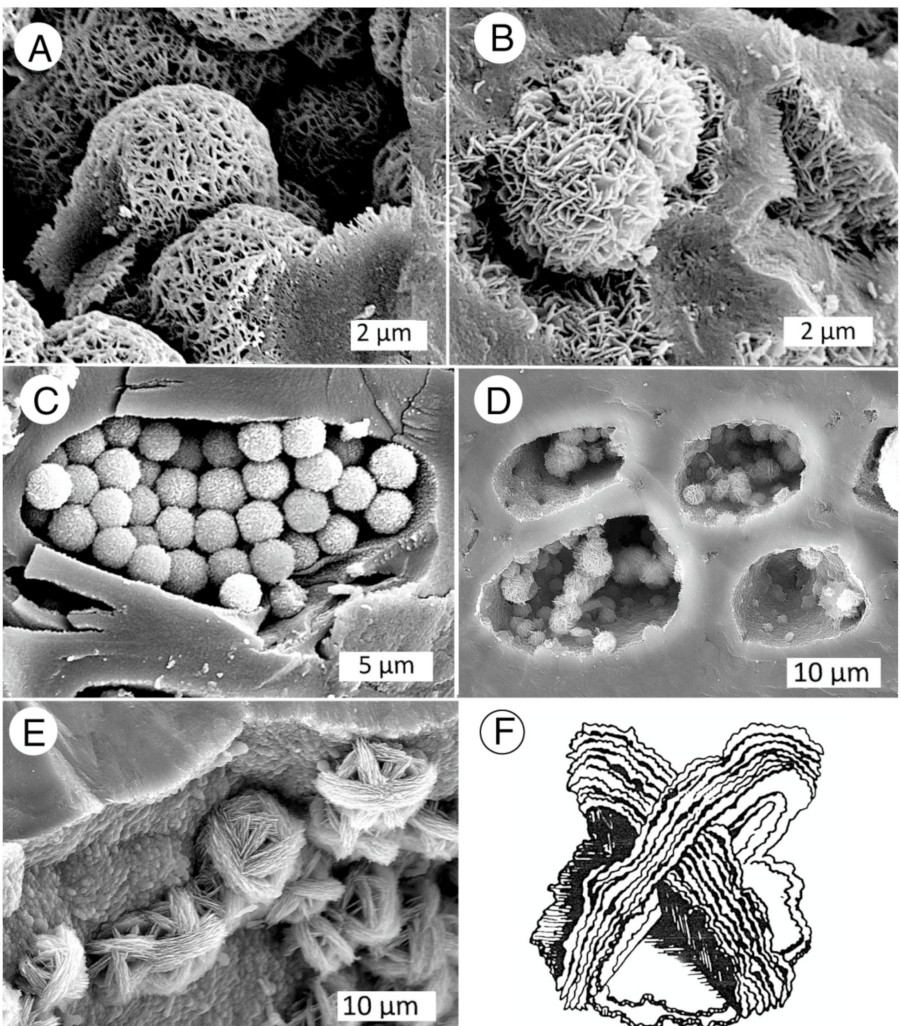

**Figure 27.** Miocene examples of opal-CT lepispheres. (**A**,**B**) Pandora Opal Mine, Virgin Valley, Nevada, USA. (**C**) Royal Peacock Opal Mine, Virgin Valley, Nevada, USA. (**D**) Georgia Nation specimen provided by Mirian Makazde. (**E**) Lepispheres showing twinned structure. Selah Butte, central Washington, USA. (**F**) Sketch showing a juvenile stage of opal-CT lepisphere formation where twinning of blades occurs at 70° angles, corresponding the faces of a cristobalite octahedron. Sketch adapted from [34]. In all of these examples, crystal shapes can be seen only in areas that provided open space for crystal growth. In other regions, the opal has a vitreous form.

*9.3. Precious Opal*

"Play of color" is the defining characteristic of precious opal, caused by dispersion of light when silica nanospheres are stacked in an ordered way. Precious opal may occur as thin layers in fractures or as three-dimensional fillings in voids, but precious opal also occurs in fossil wood. Examples include Lightning Ridge, Australia, where the opal is amorphous, and Virgin Valley, Nevada, USA where opal-CT is the dominant form of silica (Figure 28).

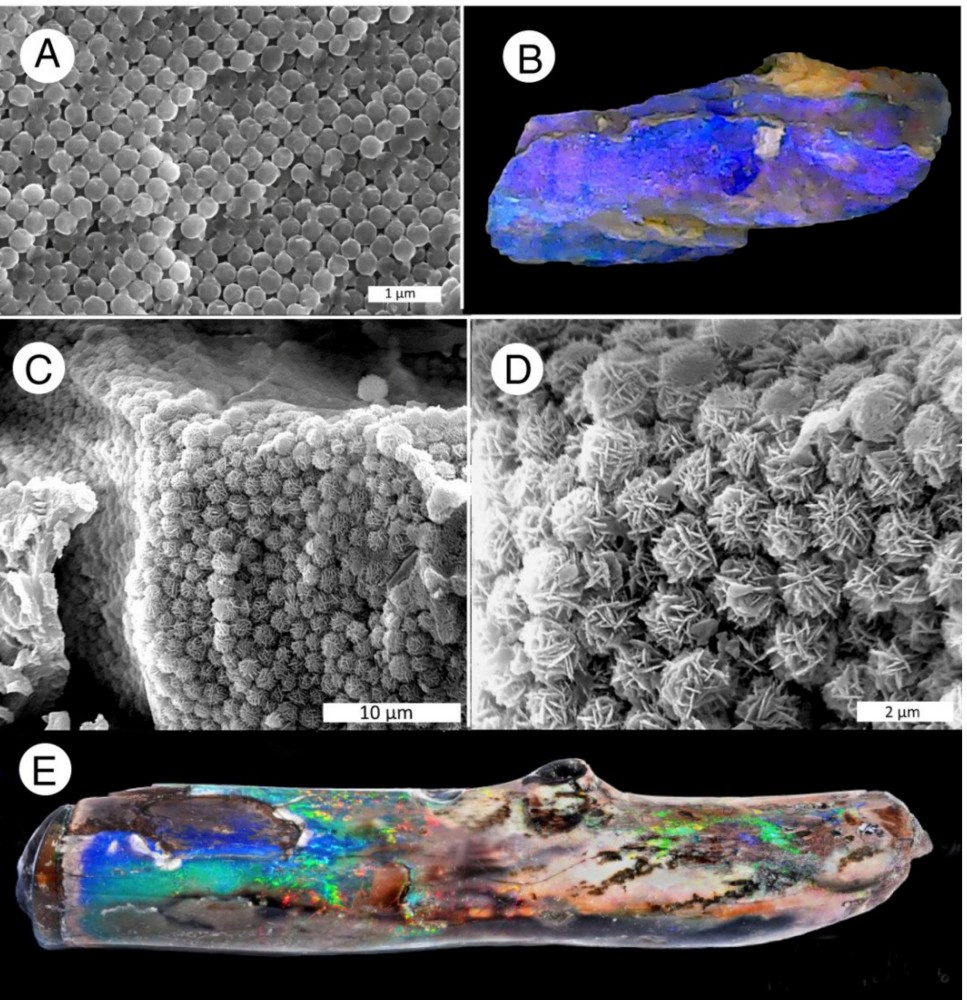

**Figure 28.** Examples of precious opal. (**A**). SEM image of Australian precious opal. Photo from Mineralogy Division, Geology and Planetary Science, Caltech University, Pasadena, California, USA, Creative Commons license. (**B**) Precious "black opal" replacing wood, Lightning Ridge. New South Wales, Australia. Specimen courtesy of Elisabeth Smith. (**C,D**) Miocene precious opal at Virgin Valley, Nevada, USA owes its play of color to the orderly arrangement of opal-CT lepispheres. Specimen provided by Richard Dayvault, (**E**) Fossil limb from Royal Peacock Mine, Virgin Valley, northern Nevada, USA.

### 9.4. Chalcedony

Chalcedony is the dominant component of silicified wood in many of the world's fossil forests, nearly universal in those of Mesozoic or Paleozoic age [108]. Chalcedony replacement is also common in Cenozoic fossil wood localities, producing specimens that are favored by collectors because the homogeneity and hardness of the silica provides excellent lapidary characteristics (Figure 29).

The fibrous or microcolumnar nature of chalcedony is commonly evident in SEM images (Figure 30).

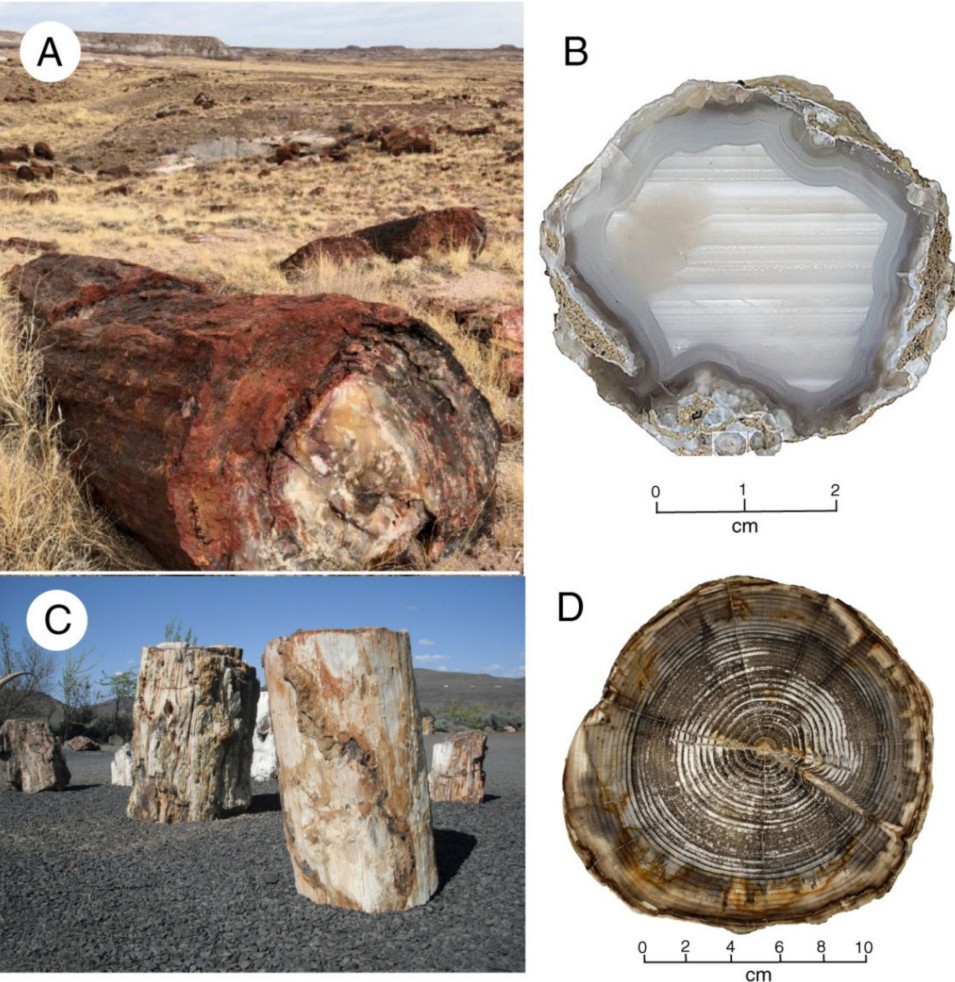

**Figure 29.** Examples of chalcedony mineralization. (**A**) Petrified Forest National Park, Arizona, USA, Triassic. (**B**) Wiggins Fork locality, near Dubois, Wyoming, USA, Eocene. This is a limb cast where chalcedony initially formed a lining layer within aa natural mold. Internal filling shows parallel geopetal layers that represent the original horizontal orientation of the wood.(**C**) Ginkgo Petrified Forest State Park, Washington, USA, Miocene. (**D**) Transverse section of *Piceoxylon* log, Saddle Mountain, central Washington, USA, Miocene. Photos (**A**,**B**) courtesy of Mike Viney, photo (**D**) by Tad Dillhoff.

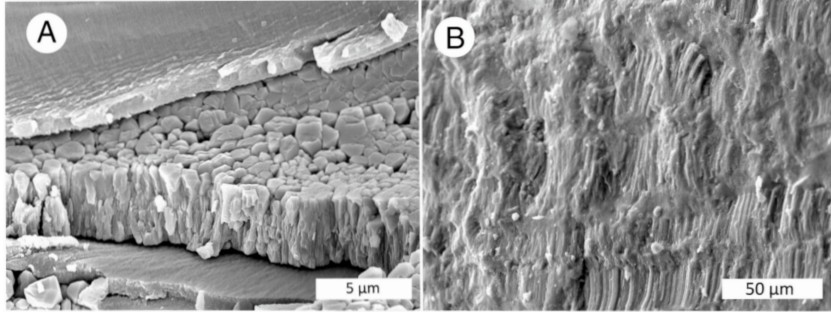

**Figure 30.** *Cont.*

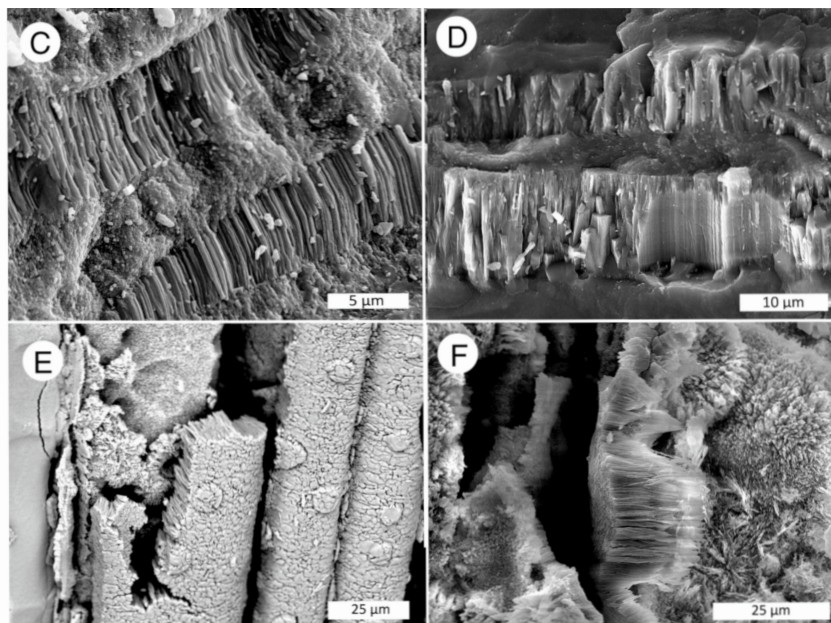

**Figure 30.** SEM views of chalcedony in Miocene fossil wood. (**A**) Blakeley Formation, northwest Washington, USA. (**B**) Northwest Iran. (**C**) Saddle Mountain, central Washington, USA. (**D**) Goose Creek, northern Nevada, USA. (**E**) Radial view showing chalcedony replacing cell wall and pit apertures. Lumen are empty. Yakima Canyon, central Washington, USA. (**F**) Fibrous chalcedony replacing cell wall. Saddle Mtn, central Washington, USA. Images (**E**,**F**) are from specimens that were treated for 60 sec with 40% HF to show mineral structure. HF treatment removes interstitial silica that would otherwise cause cell wall surfaces to appear smooth.

*9.5. Quartz*

Megacrystalline quartz is relatively uncommon in fossil wood, with the exception of voids and fractures that may contain quartz crystal linings as a late stage precipitate that developed when silica levels were relatively low, allowing time for the formation of well-ordered lattices. Less often, wood cells have been replaced by crystalline quartz. The origin remains enigmatic. One possibility is that the quartz originated from dissolution and reprecipitation of silica from a precursor phase. In other situations, quartz is perhaps a primary form of silicification, e.g., the Late Pleistocene silicified wood at Ban Tak, Thailand (Figure 31). At this locality, the direct precipitation of quartz presumably resulted from release of dissolved silica to surface waters and ground water as a result of lateritic weathering in a tropical environment [50].

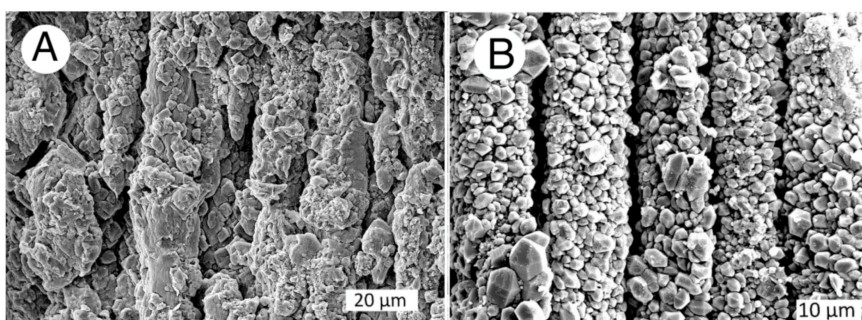

**Figure 31.** *Cont.*

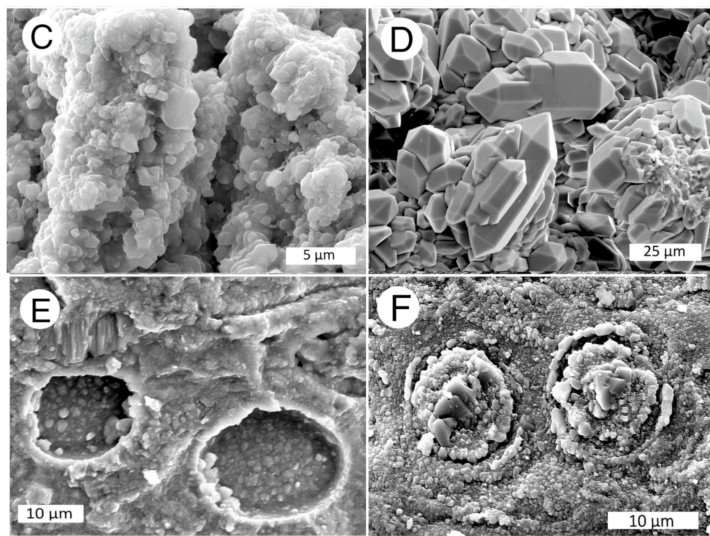

**Figure 31.** Megacrystalline quartz in fossil wood. (**A**) Longitudinal view of Late Pleistocene angiosperm wood, Ban Tak, Thailand. Specimen courtesy of Nareerat Boochai. (**B**) Longitudinal view of Cretaceous wood, Trinity Formation, Texas, USA, (**C**) High magnification view of cells replaced by quartz. Cretaceous Kirtland Formation, New Mexico, USA. (**D**) Euhedral quartz crystals in Miocene wood, Black Rock Desert near Gerlach, Nevada, USA. (**E**) Small quartz crystals lining cell lumen, Cretaceous, Saskatchewan River, Canada. (**F**) Eocene wood from Yellowstone River, Montana, USA, showing crystalline quartz as pit casts in a cell wall replaced by granular quartz. The relatively large size of the crystals tends to reduce the fidelity of preservation of cellular details, compared to chalcedony-mineralized wood.

## 10. Multiple Mineralization Phases

Wood fossilization commonly proceeds by discrete steps, where different forms of silica are precipitated in successive episode (Figure 32). This process means that the presence of several silica polymorphs may exist in a single specimen, and caution is needed before these combinations can be interpreted as diagenetic sequences, e.g., [109].

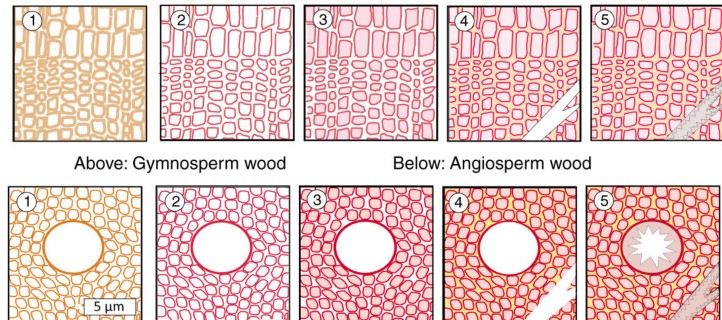

**Figure 32.** Typical mineralization sequences for angiosperms and gymnosperms, transverse views. Gymnosperms lack conductive vessels, with fluid transport occurring via tracheid cells. Angiosperms have conductive vessels surrounded by fiber cells. These anatomical characteristics affect mineralization. (**1**) Unfossilized wood. (**2**) The first step in mineralization is the deposition of silica in cell walls. (**3**) This is followed by precipitation of silica in cell lumen in fiber cells (angiosperms) and tracheids (gymnosperms). (**4**) Minerals fill intercellular spaces, producing brittleness that makes the material subject to brittle fracture if deformation occurs. (**5**) Large open spaces are the last areas to be mineralized. These spaces include vessels in angiosperm wood, and fractures or rotted areas in wood of all species. Because of the time lag between each mineralization episode, changes in geochemical conditions may cause different silica polymorphs to be deposited at each stage of mineralization. Subsequent diagenesis may alter the original mineralogy.

### 10.1. Cell-by-Cell Mineralization

Wood is composed of permeable cells that are responsible for fluid transport during the life of the plant. After death, these cells remain permeable, allowing entry of mineral-bearing groundwater. One consistent feature is that the cell walls are the first tissues to be mineralized because of organic templating. Subsequently, cell lumen may be subject to precipitation of minerals, but by then the geochemical conditions may have changed. One possibility is that the dissolved element composition of groundwater is different. Degradation of organic matter may have produced Eh or pH changes. As a result, lumen fill material is often dissimilar to the mineralization of the cell walls. Intercellular spaces and vessels may be mineralized during yet another phase where geochemical conditions have again changed. Precipitation of minerals in larger spaces (e.g., fractures or decayed regions) typically represents the final phase of mineral precipitation. These steps are discussed in detail in the following section of this paper.

In the simplest instances, cell lumen are filled in a uniform manner (Figure 33). In some specimens, lumen mineralization is more complex. Individual cells may have unique characteristics. Permeability may be inhibited by physical damage to the cell, or from air gaps that prevent fluid transport. These phenomena may prevent entry of groundwater, so the cell lumen remains unmineralized. In cells where mineralization occurs, pH and Eh maybe highly localized. The result of these phenomena is that differing mineral assemblages may occur within a single specimen (Figures 34 and 35).

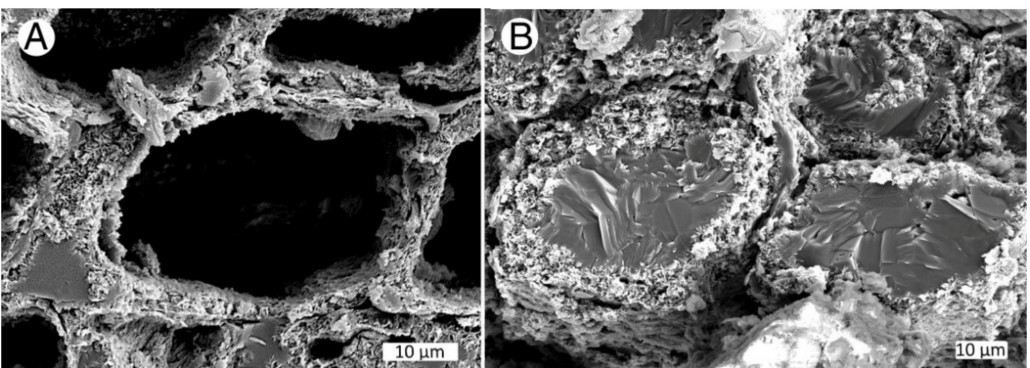

**Figure 33.** Miocene wood from Rainbow Ridge opal mine, Virgin Valley, northern Nevada, USA. (**A**) Organic templating causes deposition of amorphous opal in cell walls. (**B**) Lumen were later filled with solid opal, intercellular spaces remained open.

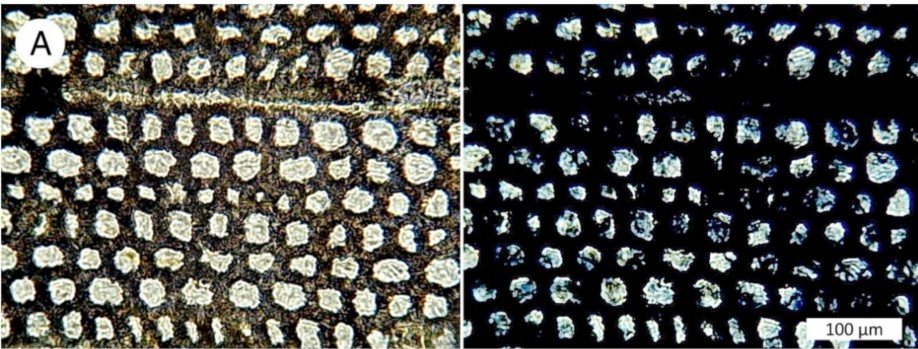

**Figure 34.** *Cont*.

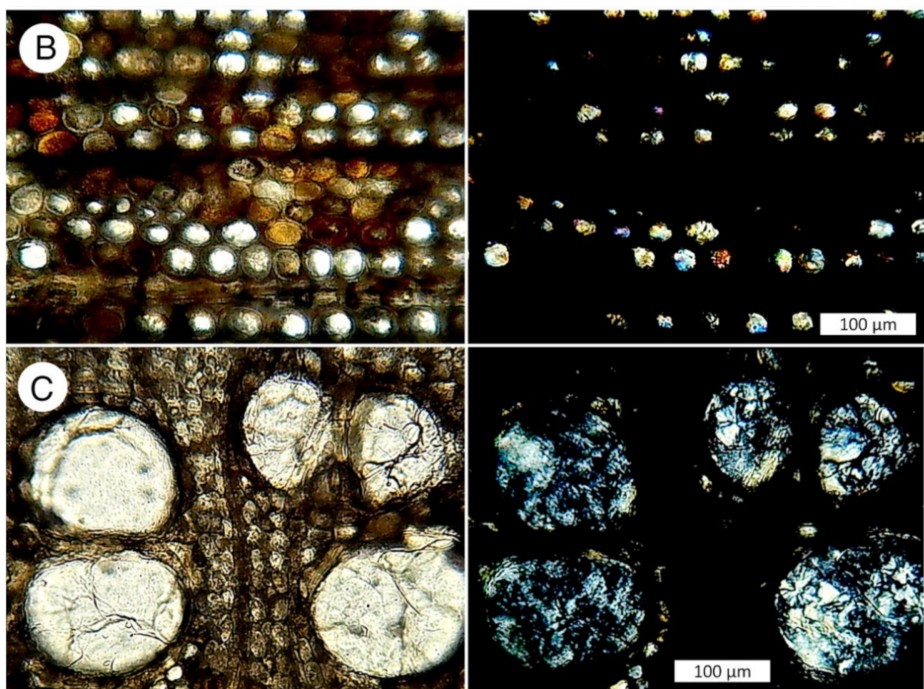

**Figure 34.** Photo pairs showing coexistence of multiple silica phases. In each pair, the left view shows ordinary transmitted light illumination, (**B**) is the same field of view under polarized light. Opal-A is identified by its opacity under polarized light. (**A,B**) Neogene fossil woods from Guatemala [110]. (**A**) Transverse view, showing cell walls mineralized with opal-A, and chalcedony filling cell lumen. (**B**) In another area on the same thin section, many cell lumen are filled with opal-A, brown in ordinary illumination, opaque in polarized light). Other neighboring cells contain chalcedony. (**C**) Miocene angiosperm wood from Yakima Canyon, central Washington, USA. Small cells are mineralized with opal; vessels contain chalcedony. Specimen courtesy of Tad Dillhoff [111].

In some locations, the composition of fossil logs is uniform (e.g., Petrified Forest, Arizona, USA), but it is not unusual for a single locality to contain specimens that have different mineralogy. Color variations may result from the presence of several mineral phases, but more commonly the colors are caused by the presence of trace elements within a single silica phase [58].

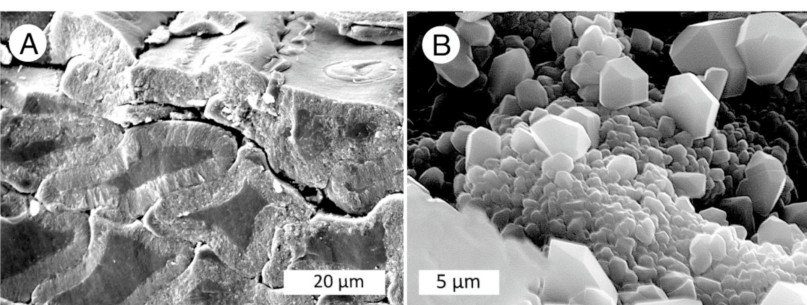

**Figure 35.** *Cont.*

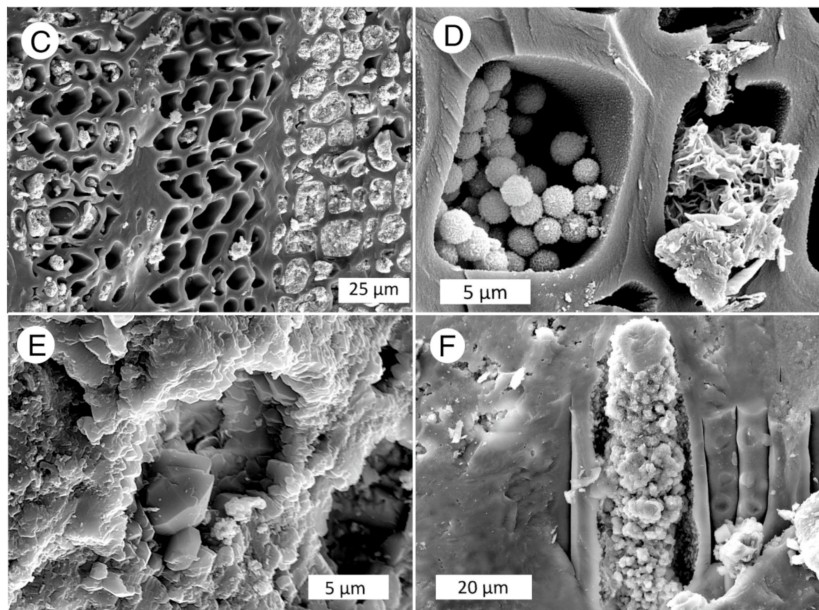

**Figure 35.** Fossil wood showing precipitation of silica in open spaces. (**A**) Eocene *Sequioxylon* wood showing wood fibers with thick cell walls, with lumen filled with fine-grained silica. Florissant Fossil Beds National Monument, Colorado, USA. Specimen provided by Herbert Meyer. (**B**) Miocene wood from Gabbs, Nevada, USA. Scattered euhedral quartz crystals have precipitated on a cell wall that was replaced by opal-A. (**C**,**D**) Fossil wood from Royal Peacock Mine, Virgin Valley, northern Nevada, USA. Specimens provided by John Church. (**C**) Transverse view shows three annual rings in carbonized wood. Cells in the central zone are mostly unmineralized. Some cells in the left layer contain clay minerals. Cells in the zone to the right are mostly filled with porous opal. (**D**) High magnification view of two cells, one containing opal-CT lepispheres, the other partially filled with a clay mineral. (**E**) Cell with fibrous chalcedony replacing cell walls, with euhedral quartz in lumen. From Miocene wood from Gardinier Phosphate Mine in Polk County, Florida, USA. Specimen provided by Harry Miller. (**F**) Crystalline quartz filling a cell lumina in opalized Miocene wood from Terrace Heights, near Yakima, Washington, USA. Specimen provided by T.A. Dillhoff.

## 10.2. Mineralization of Open Spaces

Although cell walls may experience incipient mineralization from organic templating, and silica may precipitate in open cell lumen, complete wood petrifaction requires a reduction in tissue volume. Open spaces may be produced prior to the onset of fossilization because of tissue shrinkage [112]. Decay may create voids that provide space for mineral deposition. These relatively large spaces are unlikely to be mineralized as an early stage of mineralization, remaining open during the times when smaller spaces are being filled. When wood has been initially silicified (e.g., silicification of cell walls and lumen, with intercellular spaces remaining open), the fossil wood may retain the splitting characteristics of the original wood. This fossil wood may readily split lengthwise along the grain causing cleavage along radial and tangential planes. As mineralization continues, filling of intercellular spaces makes the fossil susceptible to brittle fracture that may not correspond to wood gain orientation. Stresses caused by tectonic forces or compression or expansion of the sediment enclosing the wood can cause cracks that become potential sites for precipitation of minerals from groundwater (Figure 36).

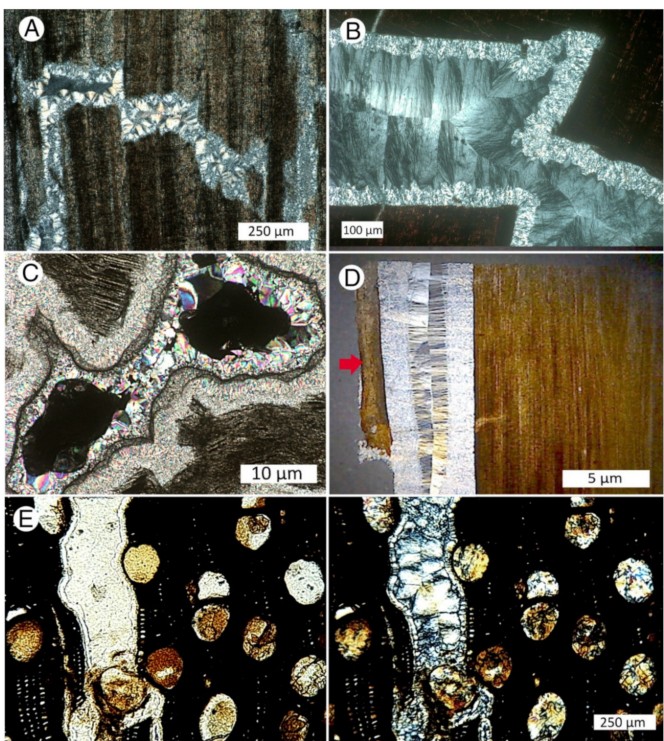

**Figure 36.** Fossilized wood containing open spaces that become sites for silica precipitation after the adjacent cellular tissue had been mineralized. (**A**) Polarized light view of a fracture containing chalcedony in Eocene wood, Yellowstone River, Montana, USA. The open fracture was offset prior to chalcedony formation, evidence that the surrounding mineralized wood had reached a stage of brittleness. (**B**) Polarized light view of an offset fracture in Eocene fossil wood from Blue Forest, Eden Valley area, Wyoming, USA. Silica precipitation in the fracture occurred in two stages, beginning with a thin microcrystalline layer coating the fracture walls, followed by formation of fibrous chalcedony that filled the remaining space. This chalcedony grew inward from the fracture walls, as evidenced by a central midline where the two layers met. (**C**) Polarized light image of a decayed region in Miocene fossil wood from Goose Creek, northern Nevada, USA. The cavity wall has a thick layer chalcedony covered by quartz crystals. The cavity center area remains unmineralized. (**D**) Chalcedony fills an open space in Miocene wood where desiccation caused the bark (marked with arrow) to separate from the interior wood. Eocene Blue Forest locality, Eden Valley area, Wyoming, USA. Polarized light view. Specimen provided by Mike Viney. (**E**) Thin section view of Miocene wood from Georgia nation Transverse view shows vessels mineralized with opal-CT, and a fracture that contains two generations of chalcedony. Polarized light view on right. Specimen provided by Mirian Makazde.

*10.3. Recrystallization*

Silicified wood may show textures caused by later recrystallization of the original silica. During the original mineralization, each cell commonly has unique patterns of mineralization, where the cell walls confine the precipitation of silica. The development of zones where the silica in many adjacent sells have identical optical properties is probable evidence of diagenetic recrystallization (Figure 37A).Likewise, the development of quartz crystal that cross-cut anatomical structure is a result of recrystallization (Figure 37B).

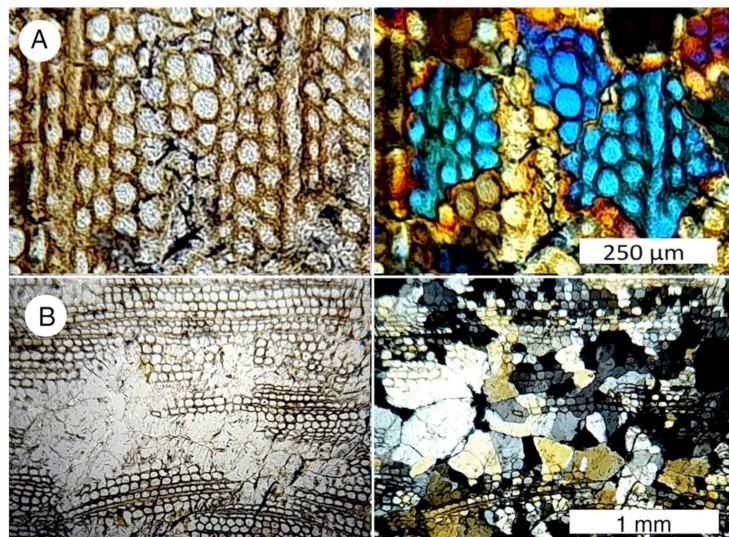

**Figure 37.** Transverse thin section views of Cretaceous silicified wood from Thailand. Polarized light image on right, ordinary transmitted light image on left. (**A**) Polarized light image shows that quartz occurs in crystallites that contain many cells. Bright birefringence colors are visible because the thin section is thicker than the usual 30 μm. This color pattern is a contrast to typical silicified woods where cells were individually mineralized. Specimen from Nakhon Ratchasima Province. (**B**) In this fossil wood, mineralized cells were later disrupted by the expansive growth of quartz crystals. Specimen from Phichit Province. Specimens were provided by Yupa Thasod.

## 11. Accessory Minerals

Silicified wood may contain a diverse variety of non-silicate minerals that may have formed contemporaneously with silicification, or as a later stage of mineralization. Accessory minerals can provide important clues for understanding the fossilization process (Figures 38 and 39).

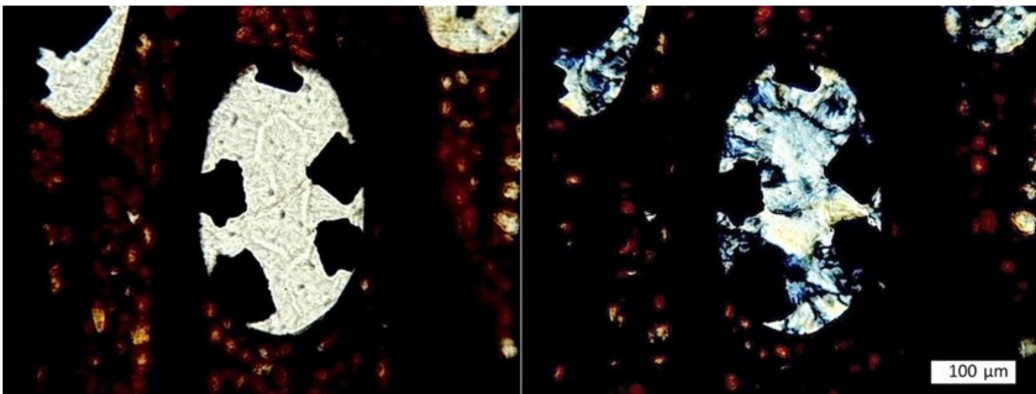

**Figure 38.** Opalized Miocene wood from Georgia nation contains pyrite crystals that are attached to the vessel walls, which subsequently became filled with chalcedony. Specimen provided by Mirian Makazde.

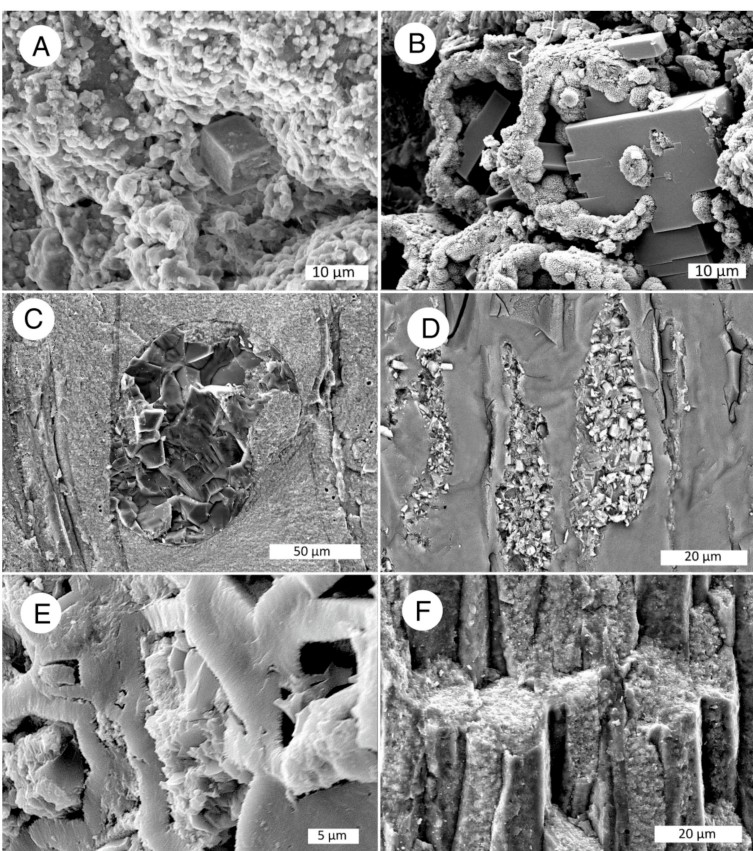

**Figure 39.** SEM images of accessory minerals. (**A**) Cubic pyrite crystal in Cretaceous wood from Texas, USA appears to have been formed contemporaneously with silicification of the adjacent cells. (**B**) Transverse view of Miocene wood from Royal Peacock Mine, Virgin Valley, Nevada, USA shows cell walls mineralized with opal-CT that is intergrown with zeolite crystals, which formed from elements released by dissolution of the tephra-rich matrix. Zeolite only occurs near the specimen end grain surface, indicating shallow penetration of the zeolite parent elements. (**C**) A vessel in silicified Paleocene wood from Cherokee Ranch, Colorado, USA contains calcite crystals formed during late-stage mineralization. (**D**) Cell lumen in Miocene wood from northwest Iran are filled with apatite microcrystals. (**E**) Cell lumen in another specimen from the same Iran locality are filled with gypsum. (**F**) Oblique radial view of Paleocene wood from Cherokee Ranch, Colorado, USA showing cells filled with iron oxides.

## 12. Diagenetic Alteration

Diagenetic transformations of silica phases are commonly inferred to occur in silicified wood, but the evidence is largely circumstantial. Opal wood is very abundant in Cenozoic fossil wood, but absent in Mesozoic and Paleozoic formations. The main apparent exception is Cretaceous sediments of Australia, where opal-A is the dominant constituent. As discussed later in this report, this opal precipitation may be the result of Cenozoic mineralization.

Transformations are well-documented for silica minerals in siliceous sinters and sediments that contain biogenic silica. In those geologic environments, opal-A alters to opal-CT, which in turn transforms to quartz. The diagenetic transformation sequence has been described in detail by Liesegang and Tomaschek [113].

The occurrence of opal-A in young deposits suggests that this material may transform to opal-CT at a rapid rate. Transformation of opal-CT to crystalline silica has been reported for Cenozoic diatomite [30–33,114–120], and confirmed in laboratory experiments [121,122].

Rates of transformation of silica polymorphs have been interpreted in various ways. Mitzutani [32] estimated ages for geologic transformations based on field observations

and experimental results (Figure 40A). Behl [118] depicted transformations as a function of temperature (Figure 40B). He inferred slow transformation rates, typically involving tens of millions of years for transformation of opal-CT to quartz. Data from deep seas cores [34] show geologic ages for occurrences of silica in biogenic sediments, again supporting slow transformation rates (Figure 41). Estimates based of siliceous sinter provide a very different perspective. Siliceous sinter is initially precipitated as opal-A, transforming with time to other forms of silica [123–126]. Herdianita et al. [126] reported that opal-A remains relatively unchanged for ~10,000 years. Between 10,000 and 50,000 years, the sinter transforms to opal-C or opal-CT; after ~50,000 years, microcrystalline quartz becomes the dominant constituent Lynne et al. [124] observed that at Roosevelt Hot Springs, Utah, USA, the transformation of amorphous silica to opal-CT and then to quartz was observed to have occurred within 1900 years.

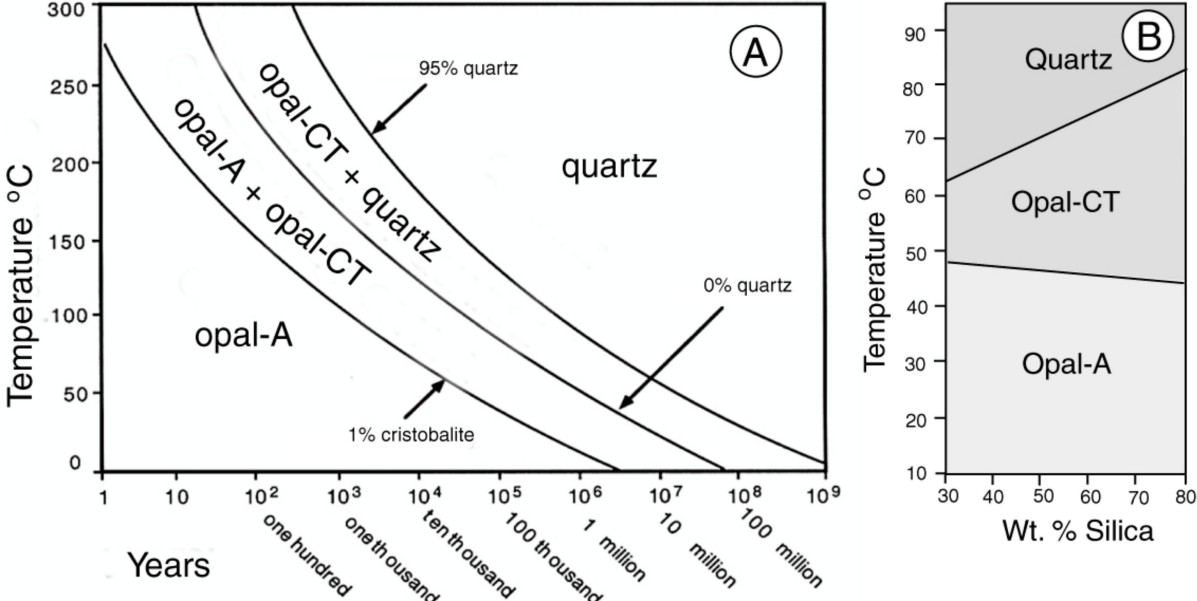

**Figure 40.** Generalized stability fields for opal and quartz. (**A**) Mitzutani [32] based his diagram on field observations and experimental studies, estimating ages required for silica phase transformations. (**B**) Behl's plot [118] is based on the % of silica in biogenic sediments versus burial temperature. Both illustrations have been adapted from the original graphics.

In natural environments, silica transformations may be affected by a variety of factors, including the presence of detrital minerals [119]. Transformation rates may be greatly accelerated in the presence of sulfur, calcite, alunite, or plant remains. For example, dissolved magnesium may accelerate the transformation because colloidal magnesium hydroxide serves as a nucleus for opal-CT crystallization [127,128]. Under some sedimentary conditions, opal-A may transform directly to quartz, without an opal-CT intermediate stage [119].

As an additional complexity for the diagenesis hypothesis, the transition of opal-A to opal-CT may not be the only explanation for the existence of these polymorphs. Both polymorphs occur in a palygorsite clay deposit in China, but opal-CT is not a crystallization product of opal-A. Opal-A precipitates during times of dry climate where groundwater input is low. Opal-CT precipitates during periods of humid climate when groundwater input is high [129].

Despite their differences, these graphs show several common trends. Although transformation rates vary, opal-A is believed to be a transitory phase that is quickly replaced by opal-C or (more commonly) opal-CT. Ultimately, microcrystalline quartz is produced from prolonged diagenesis.

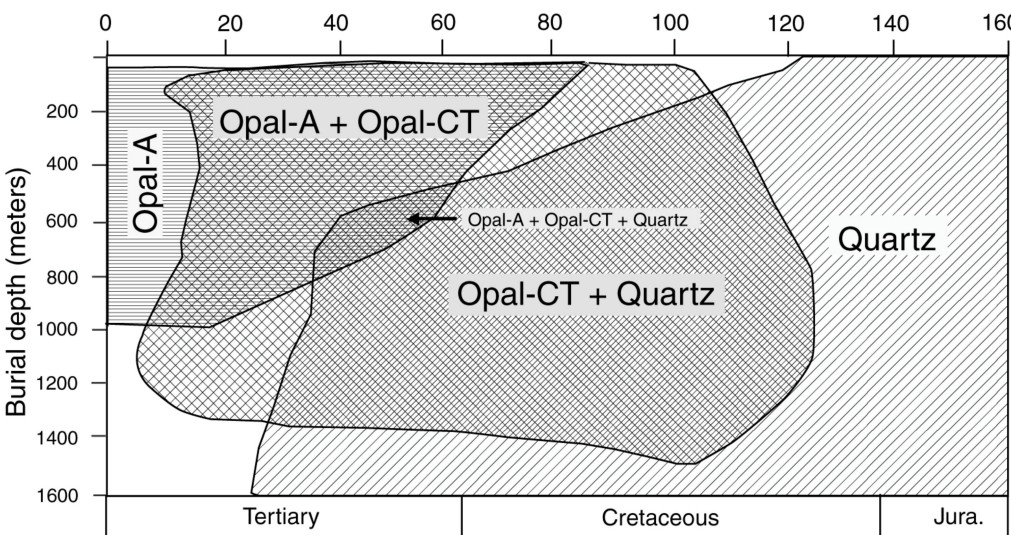

**Figure 41.** Worldwide distribution of silica polymorphs in deep sea sediments graphed with respect to geologic age. Graph adapted from Hesse [34]. Pure opal-A was only found in young Tertiary sediments. Diagenetic transformations of opal to quartz were inferred to be relatively slow. The broad opal-CT + quartz zone contrasts to continental deposits, where opal is generally absent in Mesozoic sedimentary environments. Perhaps reaction rates are very different for deep seafloor environments.

There are several issues that arise when this model is applied to silicified wood. As previously discussed, fossil wood is commonly the result of multiple stages of silicification, where various forms of silica are produced as a result of changes in the geochemical environment. The coexistence of silica polymorphs is therefore not necessarily evidence of diagenetic transformations. A second concern is that given the worldwide abundance of fossil wood localities spanning a wide age range, finding samples that are midway in the transformation process would seem to be easy. For example, fast transformation of opal-A to opal-CT might be difficult to substantiate, but the transformation of opal-CT to quartz is a more gradual transition. Why do we not commonly find specimens that have intermediate compositions (e.g., an intimate intermixture of opal and chalcedony)? Instead, samples that contain both minerals have typically resulted from independent precipitation of the two phases. Examples include the Eocene Florissant Fossil Forest in Colorado, USA [51,52] and Miocene fossil forests in central Washington, USA [111]. In some samples, opal and chalcedony occur in different regions within a single specimen, where they form zonation patterns. In other instances, individual cells contain hemispherical masses of opal enclosed within microcrystalline quartz (Figure 42).

Opal, chalcedony, and euhedral quartz sometimes occur together in associations that clearly do not involve diagenetic transformations. Alternating layers of silica polymorphs are commonly observed in veins and geodes, and in open spaces within silicified wood (Figure 43).

The fossil record lacks specimens where diagenetic transitions of silica are directly observable. The clearest evidence of diagenetic transformations comes from SEM images that preserve silica textures inherited from an earlier parent material (Figure 44).

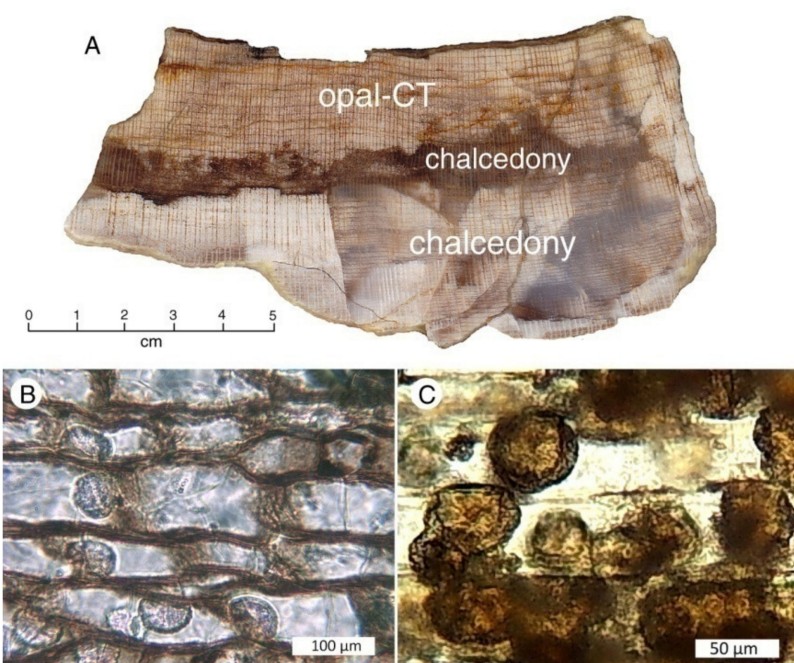

**Figure 42.** Coexistence of opal-CT and chalcedony in Cenozoic woods. (**A**) Miocene oak from Vantage, central Washington, USA. Opal-CT and chalcedony are present as separate zones. (**B,C**) Fossil wood where individual cells contain spherical masses of opal surrounded by chalcedony. These textures are evidence that the opal was not destroyed when the cells later became filled with microcrystalline silica. (**B**) *Sequioxylon*, late Eocene, Florissant Fossil Beds National Monument, Colorado, USA. Specimen courtesy of Herbert Meyer. (**C**) Miocene wood from Saddle Mountain, central Washington, USA. Specimen from Tad Dillhoff.

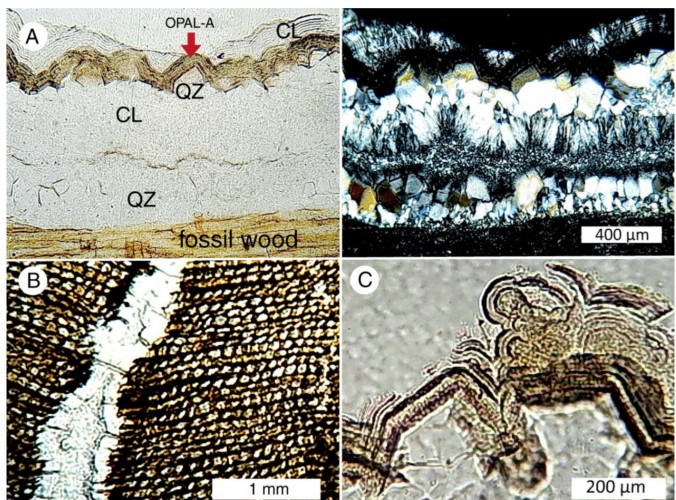

**Figure 43.** Fracture in Miocene opalized wood specimen from central Washington, USA shows layering of silica minerals deposited by successive episodes of groundwater flow. (**A**) Ordinary transmitted light image on left, polarized light view on right. Crystalline quartz (QZ) was deposited on fossil wood adjacent to the fracture. Chalcedony (CL) was deposited, followed by a second layer of quartz crystals. Terminations of these quartz crystals are coated by thin laminations of opal-A, which are covered by laminar chalcedony. (**B**) Low magnification transverse view of vein crosscutting the cell layers (**C**) Detailed view shows that opal-A layers precipitated on quartz crystal terminations have both laminar and botryoidal morphology. The precipitation of opal over quartz is not a universal phenomenon; in samples from some other localities, quartz crystals developed on an opal substrate, evidence that there is not a single geochemical pathway for silicification.

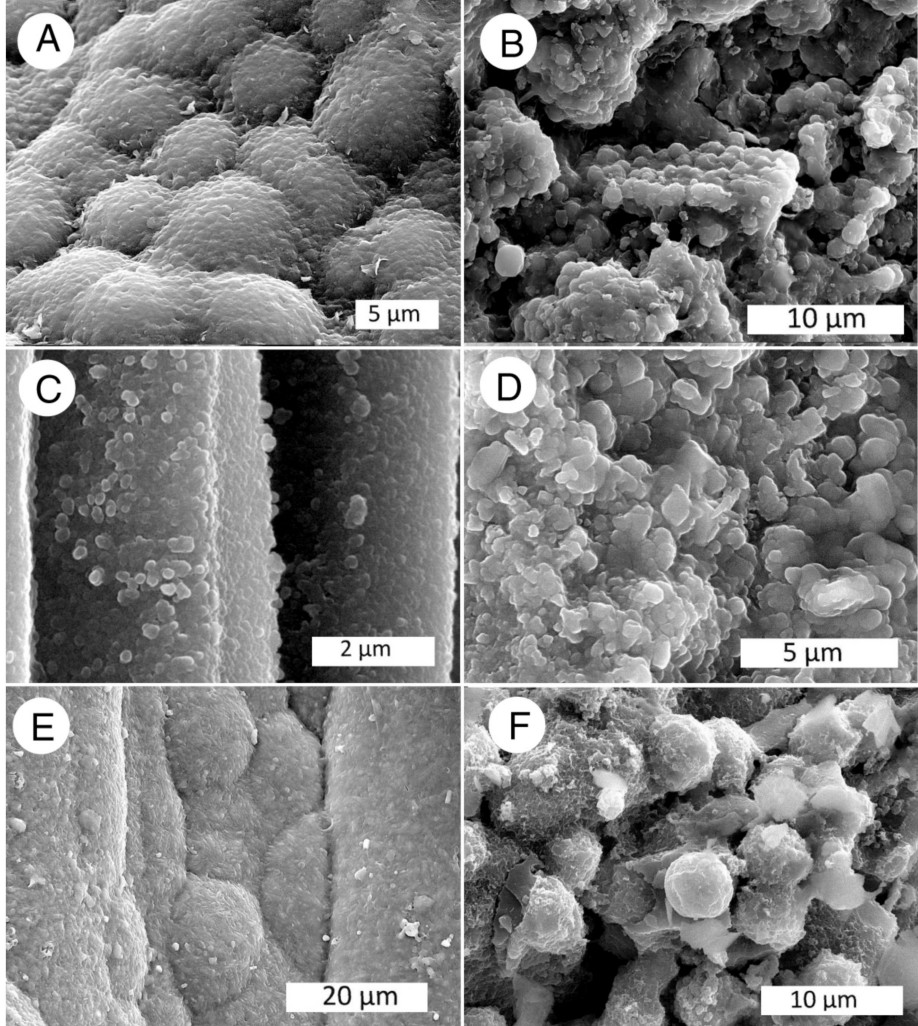

**Figure 44.** SEM images showing relict textures in silicified wood. (**A**,**B**) Chalcedony-mineralized *Protoyucca* wood showing relict opal lepisphere morphology, Miocene, Badger Flat, northeastern Nevada, USA. Specimen from Jim Mills. (**C**) Tangential view of chalcedony wood showing relict opal texture, Eocene, Blue Forest, Eden Valley, Wyoming, USA. Specimen from Mike Viney. (**D**) Relict opal textures in chalcedony wood, Eocene, Yellowstone River, Montana, USA. (**E**) Chalcedony wood showing relict botryoidal opal-CT morphology, Miocene, northwestern Iran. Specimen from Nasrollah Abbassi. (**F**) Relict opal-CT lepispheres in chalcedony wood, Miocene, Stonewall Pass, central Nevada, USA. Specimen from Patricia Caplette.

## 13. Exceptional Occurrences

The study of wood silicification has included two topics that need clarification. One concern is the common belief that silicification of wood in modern hot springs provides a model for petrifaction of ancient wood. The other issue involves the alleged presence of opal-A in the Cretaceous opal mining districts of Australia.

### 13.1. Hot Spring Evidence

Silicification may occur very rapidly when wood is exposed to the silica-rich water in hot springs [81,130–132]. For several reasons, hot spring silicification provides an unlikely model for wood fossilization in other environments. Extremely high silica levels in geothermal waters result in rapid precipitation of mineral coating on the exterior surface of wood immersed in spring water [81]. Silica may be deposited along fractures, but otherwise interior cells mostly remain unmineralized (Figure 45). Highly oxidizing conditions of alkaline chloride hot springs cause rapid degradation of organic materials, so the end

products are typically siliceous casts of wood cells rather than true petrifactions. The situation is somewhat different for standing trees, where silica-rich fluid travels upward through porous tissue [132], but the results bear little resemblance to silicified wood from older non-geothermal deposits.

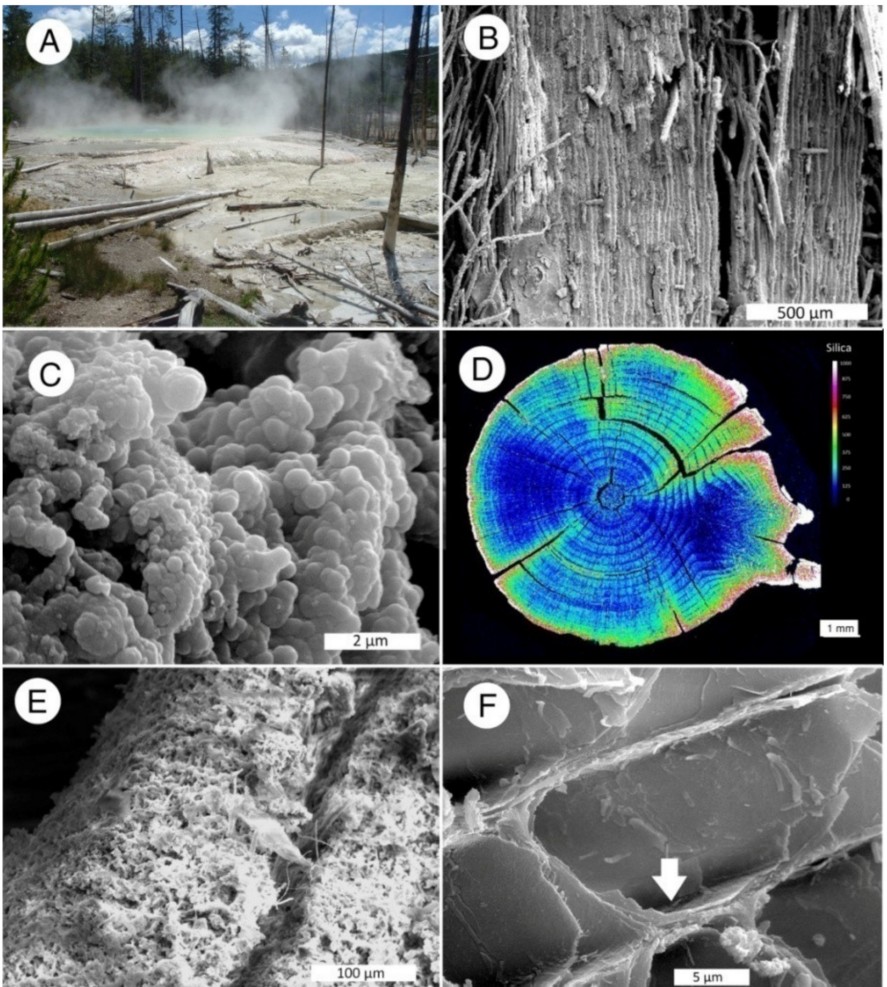

**Figure 45.** Silica precipitation in wood in modern hot springs at Yellowstone National Park, Wyoming, USA. (**A**) Cistern Spring, Norris Geyser Basin, showing abundance of lodgepole pine (*Pinus contorta*) trunks that fell into the spring, where they become encrusted with amorphous silica. (**B**) Wood cells encrusted with amorphous silica, Beehive Geyser, Upper Geyser Basin. The wood fibers were beginning to dissociate in the hot spring environment as silica was precipitating. (**C**) Opal-A lepispheres in crust enclosing a twig from Cistern Spring. (**D**) X-ray map showing silica distribution in a lodgepole pine twig from Cistern Spring. Red and orange colors show silica in the outermost zone, and penetrating along fractures [81]. (**E**) Silica crust on twig from Cistern Spring. The fibrous texture is of microbial origin; silica occurs as opal-A microspheres. (**F**) Interior of the same wood region is relatively unmineralized, with silica occurring only as incipient layers within cell walls (shown by arrow).

### 13.2. Age of Opal-A: Evidence from Australian Opal

The Cretaceous age of Australian opal deposits is known from the presence of fossils that include dinosaurs and plesiosaurs as well as invertebrates and abundant plant remains [133–135]. These fossils are composed of opal-A. This mineralogy is surprising, because opal-A is normally found only in Neogene formations [31,33].

The probable explanation for the presence of opal-A in Cretaceous strata is that amorphous silica was deposited long after the deposition of the host sediment. Ages

for this mineralization have been debated, with estimates including Late Cretaceous to Paleocene [136], continuously since Late Eocene [137], Mid- to Late Miocene [138], and Recent [139]. The petrography of the deposits has been studied in detail [140–142].

The delayed opalization is evidenced by fossils that typically consist of casts where the external morphology was molded by the find-grained sediment, creating spaces that were later filled with opal-A. Figure 46 shows small limb fragments from the Lightning Ridge opal district. The filling material is opal-A that encloses small fragments of the mudstone host sediment, but no relict plant tissue.

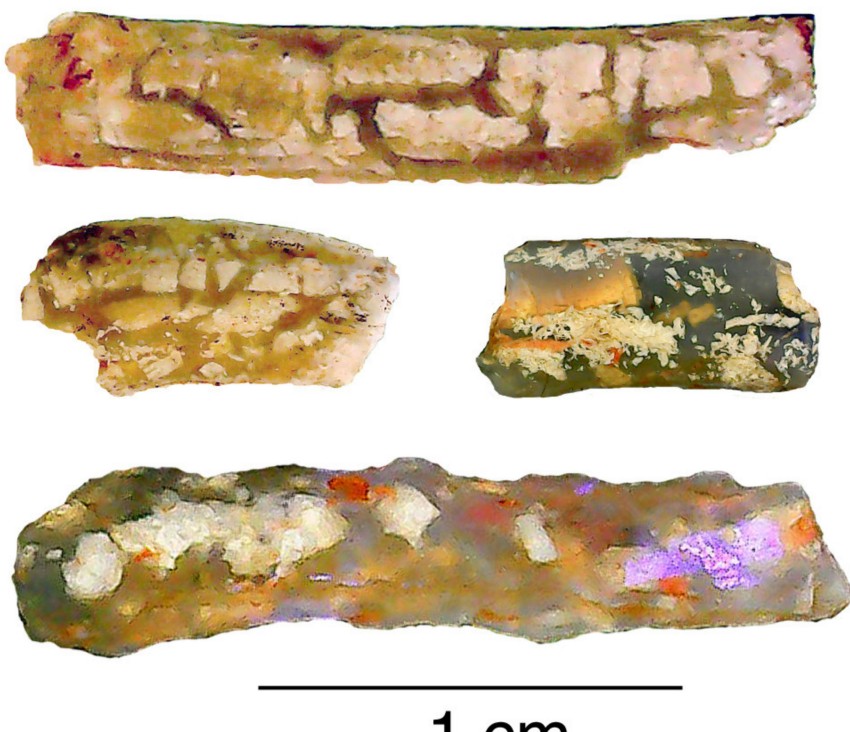

**Figure 46.** Opal-A casts of small twigs from Lightning Ridge, New South Wales, Australia. The interiors contain angular clasts of mudstone from the adjacent matrix. Specimens provided by Elisabeth Smith, Australian Opal Center at Lighting Ridge.

## 14. Conclusions

Interpretations presented in this report are a synthesis of ideas based on the author's two decades of studies of fossil wood. This work began with the author's acceptance of traditional beliefs, e.g., that silicification of wood involves permineralization, and that silica minerals typically follow a diagenetic sequence of opal-A→opal-CT→quartz [143]. Extensive examination of silicified woods from a multitude of localities revealed that, in many instances, the silicification process was variable and cannot be described by any single model. In particular, for silicified wood the permineralization hypothesis is seldom valid. More accurate nomenclature would be to describe silicified wood as being mineralized, not permineralized [106]. The latter term is best reserved only for specimens where the presence of large amounts of original organic matter can be demonstrated (e.g., by the ability to make acetate peels of silicified wood after etching the surface with HF).

The diagenetic transformation of silica is well-documented for siliceous sinter and biogenic sediments, but the existence of silica polymorphs in fossil wood can be the result of other processes. In many specimens, the coexistence of various silica minerals resulted from mineralization that occurred in multiple stages, each episode occurring under different geochemical conditions. The relative stability of silica polymorphs has been clouded by several factors. One issue is that thermodynamic studies of cristobalite and

tridymite have commonly been based on occurrences of these minerals as high-temperature phases in igneous rocks. Results of these studies reflect conditions that are very different from the geochemical environment of deposits that contain fossil wood mineralized with opal-C or opal-CT. Additional confusion arises from the common perception that Early Cretaceous sediments in Australia preserve opal-A, a mineral that may have precipitated in the Cenozoic.

Rates of silica mineral precipitation are highly variable, and it is impossible to make broad generalizations. However, the rapid rate of silica deposition in hot springs is not a reliable analog for wood petrifaction under other geologic conditions, despite the enthusiasm of Creationists for using hot spring wood to support their belief that a 6000 year age for the Earth allows adequate time for the development of all fossil forests [144,145].

The basic thesis of this report is that silicification of ancient wood can proceed via multiple pathways, sometimes involving complex processes. The study of fossil wood has long been the provenance of paleobotanists, but it is an area of research where geologists, geochemists and mineralogists can play important roles. Sophisticated analytical methods are increasingly being applied, but X-ray diffraction, polarized light microscopy, and SEM/EDS are commonly available sources of data for understanding the origin of silicified wood specimens.

**Funding:** This research received no external funding.

**Data Availability Statement:** Specimens used in this study are from the author's research collection archived at the Geology Department, Western Washington University, Bellingham, WA 98225, USA. This reference collection includes SEM stubs and thin sections.

**Acknowledgments:** The international scope of this report has been possible because of the generous access to specimens from a multitude of fossil wood localities. Figures in this report include photographs of specimens that were generously donated by Nasrollah Abbassi, Nareerat Boonchai, Patricia Caplette, John Church, Markus Eberl, Rick Dillhoff, Tad Dillhoff, Jack Hughes, David Lawler, David Lang, Mirian Makazde, Herbert Meyer, Harry Miller, Jim Mills, Elizabeth Smith, Yupa Thasod, Mike Viney and Gerald Worrell. Three reviewers deserve my thanks for their constructive suggestions for improving the manuscript.

**Conflicts of Interest:** The author declares no conflict of interest.

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
