# Peer review of "Silicification of Wood: An Overview"

_minerals, doi:10.3390/min13020206_

Round 1
Reviewer 1 Report
Well written.
Some typos need to be corrected for example in line 7 "m1nerals".
Spelling check needed. For example check the spelling of 'Chris Ballhaus' in reference.
Add few paragraphs on the kinetics of the wood silicification.
Author Response
Thanks for the encouraging review! I am hoping to fix all of the various typos in the original draft. I confess that for initial submissions I am more focused on presenting ideas than fine-tuning spelling and typing. I have added some text paragraphs and an illustration in response to your suggestion.
Reviewer 2 Report
Dear author and editor,
Please find attached my review of the manuscript 'Silicification of wood: a review' by George E. Mustoe. The paper provides a complete review of wood silicification, covering from the sources of silica to nomenclature of silica minerals, research methods from macro to microscale and the processes acting during and after silicification. The manuscript is beautifully illustrated, especially with regard to SEM images. The paper compiles the knowledge acquired by the author during his long career, providing his own ideas about the rarity of permineralization in wood silicification and its multiple-stage character. I think the manuscript is a nice piece of work, especially useful for novel researchers in this field, and that it perfectly fits with the journal topics.
I am listing here some of the more significant comments and suggestions that I think will improve the manuscript to become this main reference for geologists, in a broad sense (petrologists, geochemists, paleontologists…), who work in wood silicification. Minor comments and suggestions can be found in the annotated manuscript file. Also in the manuscript, there are many typographic errors that I have marked in the annotated version. Please do a careful revision of this.
(1) I think section 2 (Nomenclature of Silica Polymorphs) should be slightly improved and completed with other terminology that is also commonly used for opal and chalcedony, as this paper is presented as a review. On one hand, the term lussatite should be introduced. I think it is used as a synonym of opal-CT. I also miss a microphotograph of opal-CT illustrating how this incipient crystallization is observed, compared to opal-A. On the other hand, the subsection of microcrystalline quartz should be reorganized. First, you should explain why this section is called microcrystalline quartz. Then, which are these other minerals that can be found together with quartz in chalcedony. Secondly, improve terminology and description. For instance, quartzine is also known as length-slow chalcedony. Mogánite is also known as lutecite. Introduce length-fast chalcedony. If you could be able to tidy up and explain all this nomenclature we all would thank that.
(2) In my opinion, section 5 (Research methods: Laboratory analysis) would be clearer if you reorganize the different techniques according to their finality:
- Analysis to discern mineralogy (XRD, FTIR/Raman, density and LOI)
- Analysis to characterize the elemental composition (LA-ICP-MS). What about EPMA?
- Analysis to establish water content (Thermal analysis)
- Etc etc…
My area of expertise is diagenesis (of rocks) so I was shocked that common techniques (as oxygen stable isotopy) or new techniques (as absolute dating) used in diagenesis were not mentioned in this section. I was also surprised that these techniques are really absent with a few exceptions: Hehe Jiang, Cin-Ty Lee, William G. Parker (2018) Trace elements and U-Pb ages in petrified wood as indicators of paleo-hydrologic events, Chemical Geology 493, 266-280. Why do you think these analytical techniques used for instance in cherts are not applied in silicified wood?
(3) Though I understand this is a paper about silicification, not about paleobotany, maybe it would be worthy to add in the introduction a short section about the main physiologic features that can be seen in silicified wood, just a kind of a simple sketch showing cell lumen, vessels, tracheyds…and their main characteristics. This is just a suggestion about what I’d like to find in this review.
Minor comments:
Line 153-156: Absorbed or adsorbed? In line 153 you say that water in opal-C and opal-CT is absorbed, whereas in line 156 you say that opal fragility is due to high levels of adsorbed water. Are you referring to the same thing or different things? It is not clear for me. When you talk about opal in lines 154 and 155 are you referring to opal A, C, CT or all of them?
Line 254 together with figure 10 should be moved to the previous subsection (microcrystalline quartz) as they have no relation with what is explained in the current section (crystalline quartz).
Line 284: Florissant logs are Upper Eocene or Late Paleocene? There is a mismatch between text and figure caption 12A (line 306).
Line 358 “CL microscopy is potentially an important tool for studying the mineralogy of fossil wood”. I would not say that is for studying the mineralogy as CL do not help you to say if it is opal or quartz but it helps to establish the presence of different generations and the sequence of mineral precipitation. So it is more related to highlight these multiple mineralization phases that you explain in section 8.
Line 627: How do you know quartz is primary in this location? What is different from other locations? What has to be taken into account to surely say this is a primary quartz or a secondary quartz? Can you make a brief explanation? Do you have images of both cases to illustrate it?
Line 698 “open spaces may be produced prior to the onset of fossilization”. So there can be also an early-stage mineralization in these voids/fractures maybe similar to cell lumina or even at the same time?
Line 724: What textures? How to recognize them? Enumerate and make a brief description. When do they occur (i.e. high temperatures?)?
Line 737: What are the most common accessory minerals? What kind of clues do they provide? Are some of them specific from a certain environment?
Line 857 Section Hot spring evidence. I do not understand this section inside the diagenetic alteration topic as you are not talking about silica phases transformations, just about the velocity of silicification and how silicification works in this setting. So my suggestion is to create an independent section and maybe I would place it after section 8, and would name it Hot spring silicification.
Thus, after a careful review of the manuscript, I recommend publication after minor to moderate revisions.
Sincerely,
Reviewer

Author Response
First, let me say that I greatly appreciate your expertise but especially your dedication as a reviewer. I was delighted to get the detailed assessment of my manuscript. In regard to your suggestions for change, the answer is easy: I have adopted all of them. I was amused (maybe embarrassed) by your comment that I had not included any mention of XRF analysis in the manuscript. Ironically, although I generally label myself as a paleontologist, my professional career has mostly been as an analytical chemist, and I have spent many years doing XRF analysis by a variety of methods. And I've analyzed a lot of fossil wood samples by XRF. I don't know how I failed to think of that when I was writing the manuscript draft. Thanks for the careful proofreading. I confess that I never worry much about spelling or typing for first drafts, I'm more concerned about presenting ideas. And I always hop to find a reviewer like you who makes my rewrite more error-free.
Reviewer 3 Report
I agreed to review this paper because of an interest in better understanding a topic a bit outside my usual research topic. A review paper (which the title claims) should summarize the state of the art while directing the interested novice to the literature for the nitty gritty details. Instead, this paper simply makes statements with very little referencing. To give two examples (out of many), section “1.1 History of fossil wood research” includes a mere six lines (44-48) with only 5 references, all between 1972 and 1984. Surely something relevant has been published in the last 39 years! To give another example, lines 124-132 describe polymerization of silicic acid without a single reference. It is clear that the author is knowledgeable, but this is not sufficient here. The reader needs the referencing.
This paper is also difficult to evaluate because it is not clear what the author is trying to accomplish. On the one hand, the title suggests a review of the silicification of wood, but the abstract is directed more toward rejecting permineralization as the primary method for silicification. The body of the paper is equally confused, with some parts nearly a diatribe against permineralization (or recrystallization) and other parts a nice summary of the methods for studying silicification (albeit almost completely lacking appropriate references). Although the author has published a nice discussion arguing for using the term mineralization instead of permineralization (Mustoe, 2017), none of that reasoning is found here.
Given such major structural issues, I cannot complete a detailed review.
I recommend major revisions with effort directed at finding a better flow for a proper review, deletion of much of the diatribe, and the addition of sufficient references.
There are also way too many spelling and editing errors (I count four in the abstract alone).
The figures are beautiful, but the captions are way too short. Tell the reader what each image illustrates (the location is not sufficient). For a review, assume the reader has never looked at opal in the SEM!
Author Response
Thanks for reviewing my manuscript. My rewrite is incorporating some detailed suggestions from the other reviewer, and I will restrict my comments here to some general aspects that you brought up. As you mentioned, this paper is not really a "review", so I have changed the title to "Wood Silicification: an Overview".
In regard to your view that I have not included adequate references, I have been concerned about the length of my reference list, which is already at 142 citations. I feared that reviewers and editors might be concerned by that length, which is a reason I minimized the citations for the history of research section. I'm glad to report that I so far have not received any complaints about manuscript length, which gives me freedom to make some additions. Perhaps a good solution is to add some "for a complete bibliography, see...." statements, rather than list every citation. For silica chemistry and history of fossil wood list, the list of original references is very large. As for the multitude of typos, I can only say that in the face of a rather tight submission deadline, my main focus was on presenting ideas and especially on the work of preparing illustrations that used more than 30 years worth of photomicrographs (thousands of them). Thanks for the suggestion for writing more detailed captions. I will do that.
Thanks,
George
Round 2
Reviewer 3 Report
The authors have dealt with some, but by no means all, of the typographic errors. I get that time limits on submission might impact the number of typos, but these errors detract from the flow and make evaluation difficult.
For a review paper (an “overview” sure looks like the same thing to me!), a very long reference list should be expected. Pushing this off on someone else (“…for a complete bibliography, see…) does not seem a good solution. The reader is still left without adequate citations. For me personally, this is not acceptable; but it is an editorial decision rather than a reviewer decision.
The figure captions are much better.
The structural issues identified in my first review remain.
Section 8 (Wood silicification process) would seem to be a critical section but it is very short. The term “mineralization” is used but it is not clear what is meant by this. Is it replacement? Clearly the author does not think it is permineralization. On that, I do not agree that there is any assumption that if the mineral were dissolved, the original tissue would be present. When a pore space is filled, that sounds like permineralization; when silica instead preserves the organic matter, it is mostly replacement (but clearly some organic matter is instead entombed to be the dark matter seen in thin section). Silicification is a complicated process but this section does not elucidate than complexity. Section 10 addresses some of the issues but still never really defines the terms.
There are several nice summary figures (e.g., Fig. 4, Fig. 32) that lack documentation--cartoons should come at the end after full documentation of the process, not at the start. Alternatively, cite a paper where the detail IS given.
Author Response
I decline to accommodate the requirements for revision requested by Reviewer 3. These demands are divergent from the recommendations of the other two reviewers, whose suggestions I have adopted.